# Using hydrological and climatic catchment clusters to explore drivers of catchment behavior

Florian U. Jehn[1], Konrad Bestian[1], Lutz Breuer[1,2], Philipp Kraft[1], Tobias Houska[1]

[1]Institute for Landscape Ecology and Resources Management (ILR), Research Centre for BioSystems, Land Use and Nutrition (iFZ), Justus Liebig University Giessen, Heinrich-Buff-Ring 26, 35390 Giessen, Germany
[2]Centre for International Development and Environmental Research (ZEU), Justus Liebig University Giessen, Senckenbergstrasse 3, 35392 Giessen, Germany

*Correspondence to*: Florian U. Jehn (florian.u.jehn@umwelt.uni-giessen.de)

**Abstract.**

The behavior of every catchment is unique. Still, we seek for ways to classify them as this helps to improve hydrological theories. In this study, we use hydrological signatures that were recently identified as those with highest spatial predictability to clusters 643 catchments from the CAMELS data set. We describe the resulting clusters concerning their behavior, location and attributes. We then analyze the connections between the resulting clusters and the catchment attributes and relate this to the co-variability of the catchment attributes in the eastern and western US. To explore whether the observed differences result from clustering catchments by either climate or hydrological behavior, we compare the hydrological clusters to climatic ones. We find that for the overall data set climate is the most important factor for the hydrological behavior. However, depending on the location, either aridity, snow or seasonality has the largest influence. The clusters derived from the hydrological signatures partly follow eco regions in the US and can be grouped into four main behavior trends. In addition, the clusters show consistent low flow behavior, even though the hydrological signatures used describe high and mean flows only. We can also show that most of the catchments in the CAMELS dataset have a low range of hydrological behaviors, while some, more extreme catchments, derivate form that trend. In the comparison of climatic and hydrological clusters, we see that the widely used Koeppen-Geiger climate classification is not suitable to find hydrologically similar catchments. However, in comparison with a novel, hydrologically based continuous climate classifications, some clusters follow the climate classification very directly, whilst others do not. From those results, we conclude that the signal of the climatic forcing can be found more explicitly in the behavior of some catchments than in others. It remains unclear if this is caused by a higher intra-catchment variability of the climate or a higher influence of other catchment attributes, overlaying the climate signal. Our findings suggest that very different sets of catchment attributes and climate can cause very similar hydrological behavior of catchments - a sort of equifinality of the catchment response.

## 1 Introduction

Every hydrological catchment is composed of a unique combination of topography and climate, which makes their discharge heterogeneous. This, in turn, makes it hard to generalize behavior beyond individual catchments (Beven, 2000). Catchment classification is used to find patterns and laws in the heterogeneity of landscapes and climatic inputs (Sivapalan, 2003). Historically, this classification was often done by simply using geographic, administrative or physiographic considerations. However, those regions proved to be not sufficiently homogenous (Burn, 1997). Therefore, it was proposed to use seasonality

measures with physiographic and meteorological characteristics, but it was deemed difficult to obtain this information for a large number of catchments (Burn, 1997), even if only simple catchment attributes (e.g. aridity) are used (Wagener et al., 2007). Nonetheless, in the last decade datasets with hydrologic and geological data were made available, comprising information of hundreds of catchments around the world (Addor et al., 2017; Alvarez-Garreton et al., 2018; Newman et al., 2014; Schaake et al., 2006). This is a significant step forward as those large sample datasets can generate new insights, which

are impossible to obtain when only a few catchments are considered (Gupta et al., 2014). Different attributes have been used to classify groups of catchments in those kind of datasets: flow duration curve (Coopersmith et al., 2012; Yaeger et al., 2012), catchment structure (McGlynn and Seibert, 2003), hydro-climatic regions (Potter et al., 2005), function response (Sivapalan, 2005) and more recently, a variety of hydrological signatures (Kuentz et al., 2017; Sawicz et al., 2011; Toth, 2013). Quite often, climate has been identified as the most important driving factor for different hydrological behavior (Berghuijs et al.,

2014; Kuentz et al., 2017; Sawicz et al., 2011). Still, it is also noted that this does not hold true for all regions and scales (Ali et al., 2012; Singh et al., 2014; Trancoso et al., 2017). In addition, a recent large study of Addor et al. (2018) has shown that many of the hydrological signatures often used for classification, are easily affected by data uncertainties and cannot be predicted using catchment attributes. Another recent study by Kuentz et al. (2017) used an extremely large datasets of 35,000 catchments in Europe and classified them using hydrological signatures. For their classification, they used hierarchical

clustering and evaluated the result of the clustering by comparing variance between different numbers of clusters. They were able to find ten distinct classes of catchments. However, Kuentz et al. (2017) used some of the signatures identified to have a low spatial predictability by Addor et al. (2018). In addition, one third of their catchments was aggregated in one large class with no distinguishable attributes. Overall, we conclude that no large sample study exists that uses only hydrological signatures with a good spatial predictability. In addition, if the climate is the dominant driver of catchment behavior, clustering catchments

based on their hydrological behavior should result in clusters with a similar climate.

Therefore, we selected the best six hydrological signatures with spatial predictability to classify catchments of the CAMELS (Catchment Attributes and MEteorology for Large-Sample Studies) dataset (Addor et al., 2017). Those six hydrological signatures are evaluated together with the sixteen catchment attributes that were shown to have a large influence on hydrological signatures (Addor et al., 2018). The connection between the hydrological signatures and the catchment attributes

is determined by using quadratic regression of the principal components (of the hydrological signatures) and the catchment attributes. This will help to explore, if a clustering with hydrological signatures that have a high predictability in space, provides hydrologically meaningful clusters and how those are related to catchment attributes. In addition, we compare the hydrologically derived clusters with climatic clusters and determine the spatial distance between the most hydrologically similar catchments. This will determine if grouping catchments by climate or by hydrologic behavior will yield the same results

and if the signatures identified by Addor et al. (2018) as having the highest spatial predictability can be used to delineate hydrologically meaningful clusters, even though they do not consider low flows.

## 2 Material and Methods

### 2.1 Data base

This work is based on a detailed analysis of catchment attributes and information contained in hydrological signatures. The

CAMELS data set contains 671 catchment in the continental united states (Addor et al., 2017) with additional meta information such as slope and vegetation parameters. For our study, we used a selection of the available meta data. We excluded all catchments that had missing data, which left us with 643 catchments. Those catchments come from a wide spectrum of characteristics like different climatic regions, elevations ranging from 10 to almost 3,600 m a.s.l. and catchment areas ranging from 4 to almost 26,000 km². We used the following attributes per class. *Climate*: aridity, frequency of high precipitation

events, fraction of precipitation falling as snow; precipitation seasonality, *Vegetation*: forest fraction, green vegetation fraction maximum, LAI maximum; *Topography*: mean slope, mean elevation, catchment area; *Soil*: clay fraction, depth to bedrock, sand fraction; *Geology*: dominant geological class, subsurface porosity, subsurface permeability. Those catchment attributes were chosen due to their ability to improve the prediction of hydrological signatures (Addor et al., 2018) and because they are relatively easy to obtain, which will allow a transfer of this method to other groups of catchments world-wide.

Hydrological signatures cover different behaviors of catchments. However, many of the published signatures have large uncertainties (Westerberg and McMillan, 2015) and lack in predictive power (Addor et al., 2018). Therefore, we used the six hydrological signatures with the best predictability in space (Table 1) (Addor et al., 2018). Those signatures were calculated for all catchments. Due to this selection, no signatures that capture low flow behavior were used, as those signatures have a very low spatial predictability.


**Table 1: Applied hydrological signatures on the discharge data of the CAMELS data set (Addor et al., 2018).**

| Signature | Unit |
|---|---|
| Mean annual daily discharge | mm d$^{-1}$ |
| Mean winter daily discharge (Nov. – Apr.) | mm d$^{-1}$ |
| Mean half-flow date; Date on which the cumulative discharge since October first reaches half of the annual discharge | day of year |
| 95 % Flow quantile (high flow) | mm d$^{-1}$ |
| Runoff ratio | - |
| Mean summer daily discharge (May – Oct.) | mm d$^{-1}$ |

**2.2 Data analysis**

The workflow of the data analysis considers a data reduction approach with a principal component analysis and a subsequent clustering of the principal components, similar to Kuentz et al. (2017) and McManamay et al. (2014). For the principal component analysis and the clustering, we used the Python package sklearn (0.19.1). The code is available at GitHub (Jehn, 2018). Validity was checked by also clustering a random selection of 50 and 75 % of all catchments. This showed that the

clustering stayed the same, independently of the number of catchments used (not shown). In all further analysis, we used all catchments to get a sample as large as possible to be able to make statements that are more general.

*Calculation of the principal component analysis*

The principal components were calculated from the six hydrological signatures described above (Table 1). We used a principal

component analysis on the hydrological signatures to remove correlations between the single hydrological signatures. We only used principal components that together account for at least 80% of the total variance of the hydrological signatures, which resulted in two principal components. Those two principal components contain the uncorrelated information of all hydrological signatures used and thus can be seen as describers of the hydrological behavior in regard to the overall amount of discharge, its distribution throughout the year, high flows and runoff-ratio. Therefore, catchments with similar principal components have

similar hydrological behavior along those signatures.

*Evaluating the connection between the principal components and the catchment attributes*

1) First, we calculated quadratic regressions between the two principal components and the catchment attributes (with the principal component as the dependent variable). This resulted in one coefficient of determination (R²) for each

pair of principal component and catchment attribute (e.g. PC 1 and aridity).

2) We then weighted the R² by the explained variance of the principal components. This addresses the differences in the explained variance of the principal components (e.g., PC 1 explained 75% of the variance, PC 2 explained 19% of the variance).

3) The weighted coefficients of determination of the two principal components were subsequently added to obtain one coefficient of determination for every catchment attribute.

Quadratic regression was selected as interactions in natural hydrological systems are known to have unclear patterns and can therefore often not be fitted with a simple straight line (Addor et al., 2017; Costanza et al., 1993). This was done first for the whole dataset and then for all clusters separately. This procedure captures the pattern on the catchment attributes in the PCA space of the hydrological signatures (for examples of this pattern see Figure A1).

*Clustering the principal components*

The principal components of the hydrological signatures were clustered following agglomerative hierarchical clustering with ward linkage (Ward, 1963), similar to previous studies (Kuentz et al., 2017; Li et al., 2018; Yeung and Ruzzo, 2001). Therefore, the clusters are based on the hydrological signatures of the catchments. From the previous studies, Kuentz et al. (2017) provides the largest set with over 35,000 catchments. They also clustered their catchments in a PCA space of a range of hydrological signatures. To select the number of clusters, they used the elbow method (and two other methods to validate their results) and found that ten or eleven clusters (depending on the method) were most appropriate for their data. Due to the similarity in the clustered data and the larger database of Kuentz et al. (2017), we also used ten clusters. (Berghuijs et al., 2014) also found that ten clusters captured the distinct hydrological behaviors for the continental US. Those ten clusters represent groups of catchments with distinctly different hydrological behavior.

## 3 Results and Discussion

### 3.1 Catchment attribute correlations in the CAMELS data set

Usually the 100th meridian is seen as the dividing climatic line in the US, splitting the country in a semi-arid west and a humid east. We assume that this difference in climate also has implications for the hydrology and the overall catchment attributes in those regions. To quantify this we split the CAMELS data set into a western and an eastern part, based on the 100th meridian (Figure 1 and 4). This shows that many of the catchment attribute correlations do not differ much between the east and the west. In most cases (>80%), Spearman rank correlation coefficients vary by less than 0.4 (Figure 1c). Still, there are some catchment attributes with larger differences of up to 0.8 between both regions. Most striking are the mean elevation and the fraction of the precipitation falling as snow as well as the vegetation attributes LAI maximum and Green vegetation fraction

maximum. Even though these attributes are directly related to each other through temperature gradients, they differ substantially in both parts of the country. In the mountainous western US, elevation is highly correlated with the fraction of precipitation falling as snow (r=0.8), while it is not in the eastern US (r=0.4). This, and the different correlations between vegetation and elevation are probably caused by the fact that the temperature gradients differ in both regions. The western US

it is much more mountainous and thus temperatures typically change with elevation. In the more level eastern US, the change in temperature is mainly linked to the latitude. Striking are also the changes of correlation with regard to the fraction of precipitation falling as snow. Here we find altered directions of the correlation, i.e., positive correlations with LAI maximum and frequency of high precipitation events in the east turn to negative ones in the west. The change in the LAI maximum might be linked to the higher elevations in the west, as in higher elevations less vegetation is growing, but more snow falls. It also

becomes obvious that all three measures of vegetation seem to track similar characteristics in the catchments, as they highly correlate with each other (especially in the eastern US with r=0.9). In addition, all vegetation attributes depict a large negative correlation with aridity. Hence, the vegetation attributes considered are likely good proxies for aridity. Overall, we see that the relations between the catchment attributes are quite similar for the eastern and western US, with the exception of the mean elevation, snow and the LAI maximum.

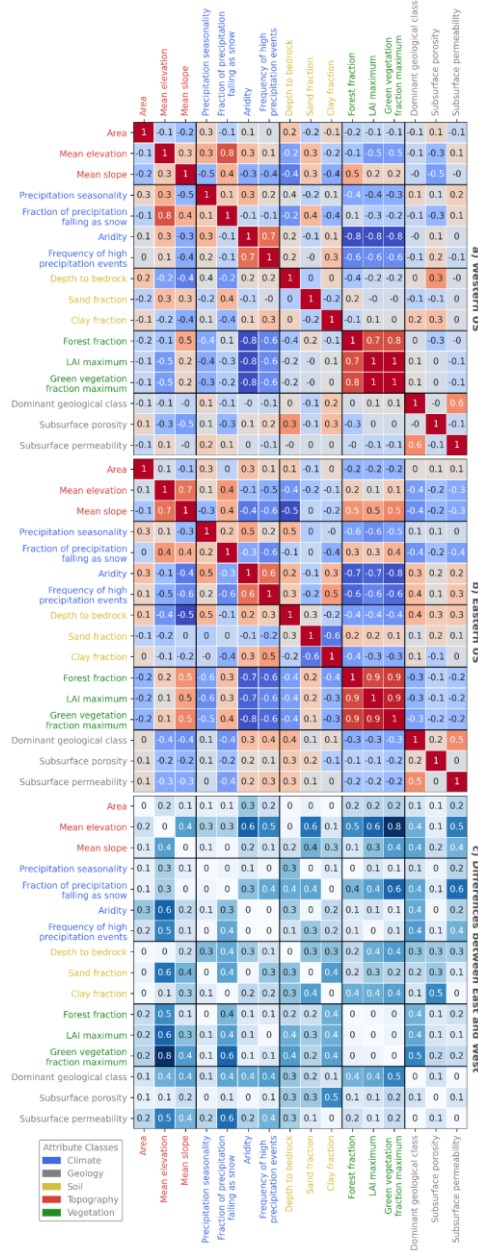


Figure 1: Spearman rank correlation coefficients given for all catchment attributes in western (a) and eastern (b) US. Absolute differences of the correlation coefficients between the eastern and western US is given in c). Eastern and western is defined by the 100[th] meridian. Due to rounding effects, correlations with the same Spearman rank correlation coefficient might show slightly varying color codes.


**3.2 Impacts of catchment attributes on discharge characteristics in the whole dataset**

Next we examined the weighted $R^2$ of the catchment attributes for the whole dataset. This analysis shows not only differences in their score between the single attributes, but also between the different classes of catchment attributes (Figure 2). Attributes related to climate (aridity) and vegetation (forest fraction) get the highest scores. However, it should be noted that all vegetation catchment attributes show a strong correlation with the aridity (Figure 1) and thus capture similar trends, in both the east and the west. With the exception of the mean slope, the first seven catchment attributes are all related to climate and vegetation. The last seven attributes on the other hand are all related to soil and geology, except the catchment area. They also show much lower scores of the weighted $R^2$. This indicates that soil and geology are less important for the chosen hydrological signatures. Similar patterns were also found by (Yaeger et al., 2012). They stated climate as the most important driver for the hydrology. As the correlations between the catchment attributes showed that the climate and the vegetation attributes are highly correlated (Figure 1), it can be assumed that climate is the overall most important factor, with aridity and high precipitation events being most important.

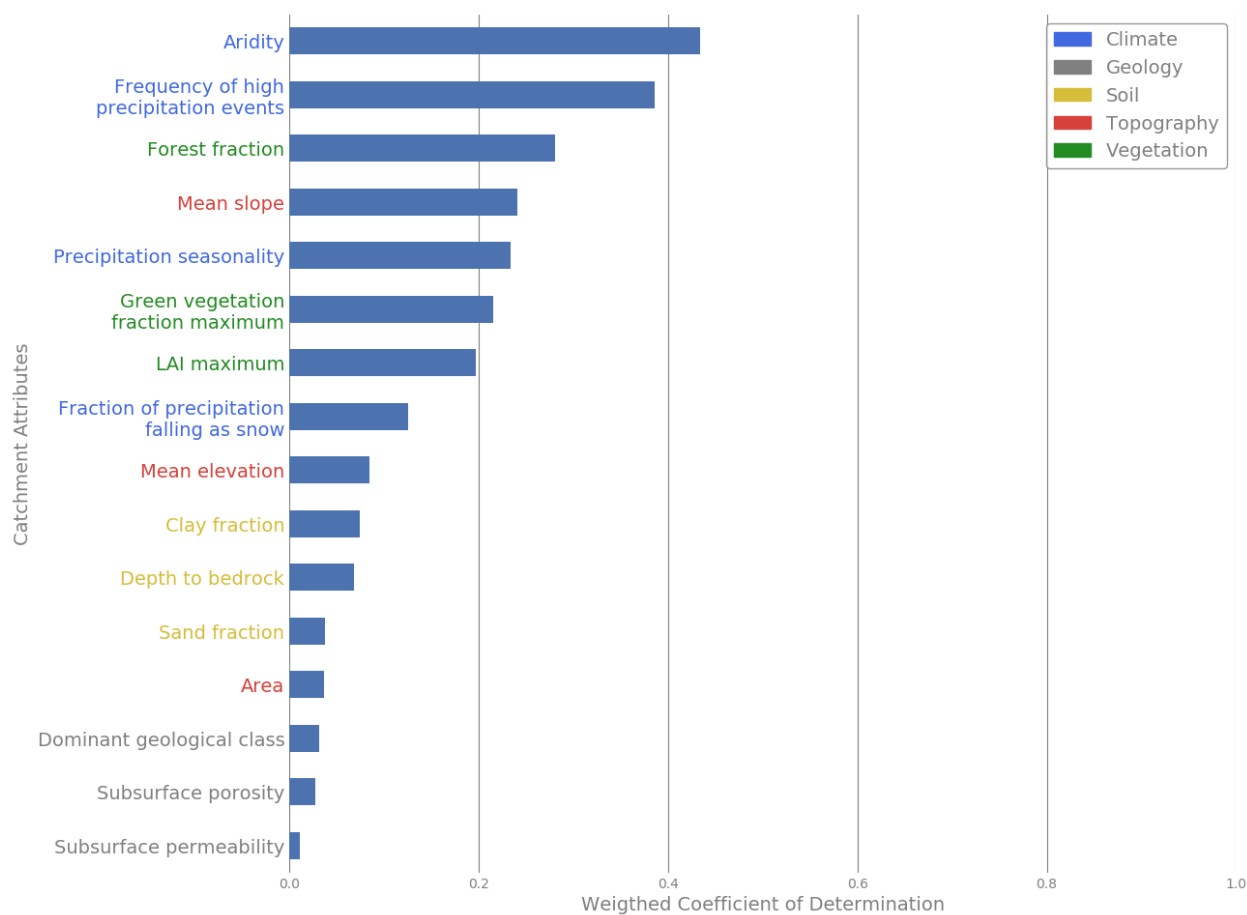

**Figure 2: Importance of catchment attributes evaluated by quadratic regression for all considered catchments. Attributes colored according to their catchment attribute class.**

However, Yaeger et al. (2012) also unraveled that low flows are mainly controlled by soil and geology. The minor importance of soil and geology in our study might therefore be biased by the choice of hydrological signatures, which excluded low flow signatures due to their low predictability in space. Nevertheless, our study probably captures a more general trend as we used a larger dataset and hydrological signatures that vary more gradually in space (Addor et al., 2018). Addor et al. (2018) also explored the influence of different catchment attributes in the CAMELS dataset on discharge characteristics. They found that climate has the largest influence on discharge characteristics, well in agreement with Coopersmith et al. (2012). The latter also used a large group of catchments in the continental United States from the MOPEX dataset. They conclude that the seasonality of the climate is the most important driver of discharge characteristics. While the seasonality is still important in our analysis, the aridity is an even stronger factor. However, Coopersmith et al. (2012) only analyzed the flow duration curve, which has a mediocre predictability in space and it is therefore less clear what it really depicts (Addor et al., 2018). Overall, this study here

is in line with other literature in the field. Using the weighted R² reliably detects climatic forcing as the most important for the discharge characteristics for a large group of catchments.

**3.3 Relation of the principal components and the hydrological signatures**

The rivers considered in this study show a wide range in hydrological signatures. This is visible in the clusters of principal components of the hydrological signatures (Figure 3). Most of the rivers are opposite of the loading vectors (the loading vectors are shown as arrows). This shows that most rivers have relatively low values for all hydrological signatures and only some, more extreme rivers, have higher values for specific hydrological signatures. Most typical for the overall behavior of the river are the hydrological signatures mean annual discharge and Q95 (high flows), as they have a strong correlation with the first

principal component. For the second principal component, the mean half-flow date has the highest correlation. Therefore, the first principal component can be seen as a measure of overall discharge and amount of high flows. Overall, it can also be seen that most of the rivers show a relatively similar behavior (cluster 1, 2, 8, 9, 10), while smaller groups of rivers tend to derivate from that by having a more extreme behavior (cluster 3, 5, 7). The remaining clusters 4 and 6 are located between those extremes. This pattern also explains the different sizes of the clusters. While most catchments behave relatively similar, only

some show extreme behavior and thus the clusters with extreme catchments are smaller.

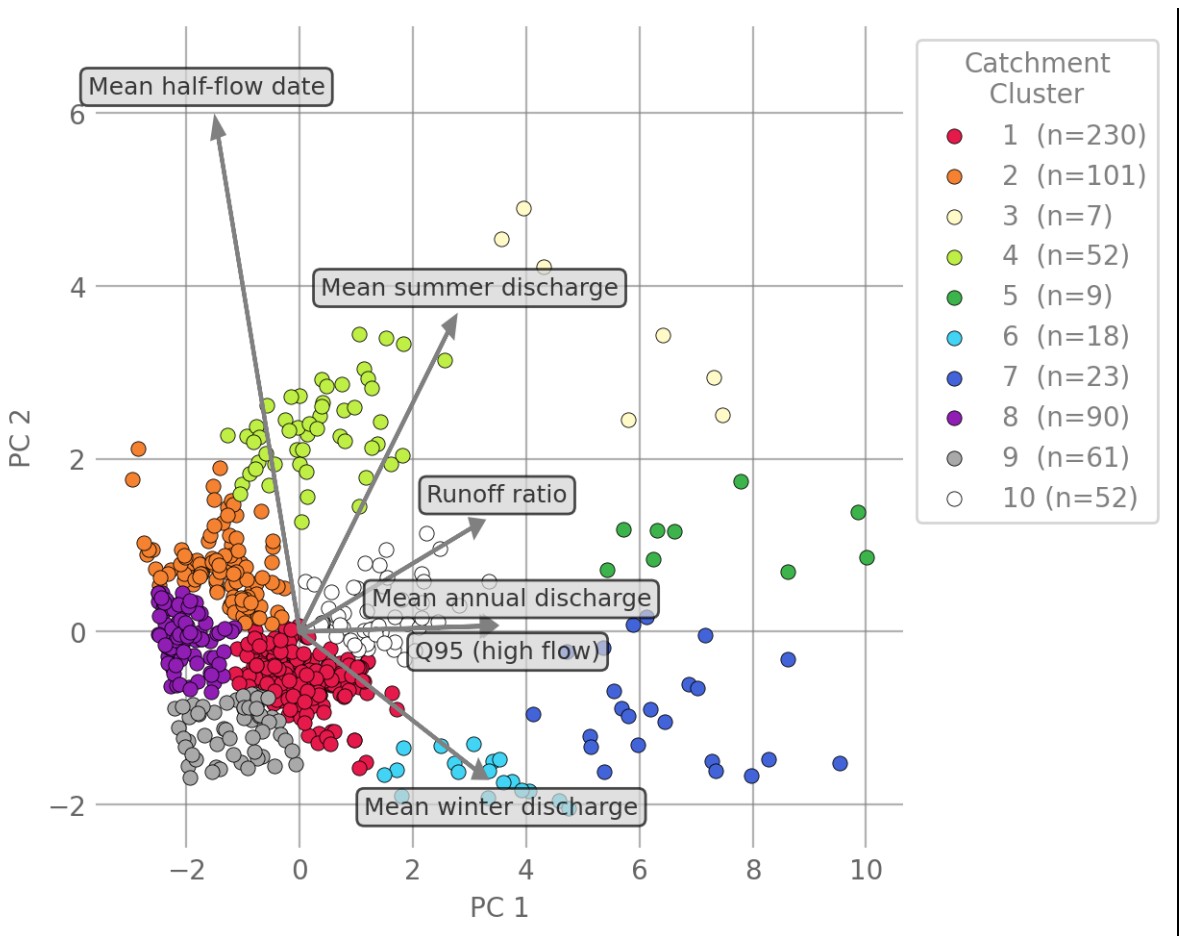

**Figure 3: Biplot of the principal components (PC). Colors indicate the cluster of the catchment. Grey arrows indicate the loadings of the original catchment attributes in the PCA space.**

## 205    3.4 Location and properties of the catchment clusters

The catchment attributes in the CAMELS and similar large scale datasets often show a pattern that resembles climatic zones (Addor et al., 2018; Coopersmith et al., 2012; Yaeger et al., 2012). For the catchments clusters presented here, we can see that most of the clusters roughly follow ecoregions in the US (Figure 4). Especially clusters 1, 4, 6 and 7 are almost entirely located within one ecoregion. Cluster 2, 8 and 9 on the other hand follow those ecological boundaries to a lesser degree.

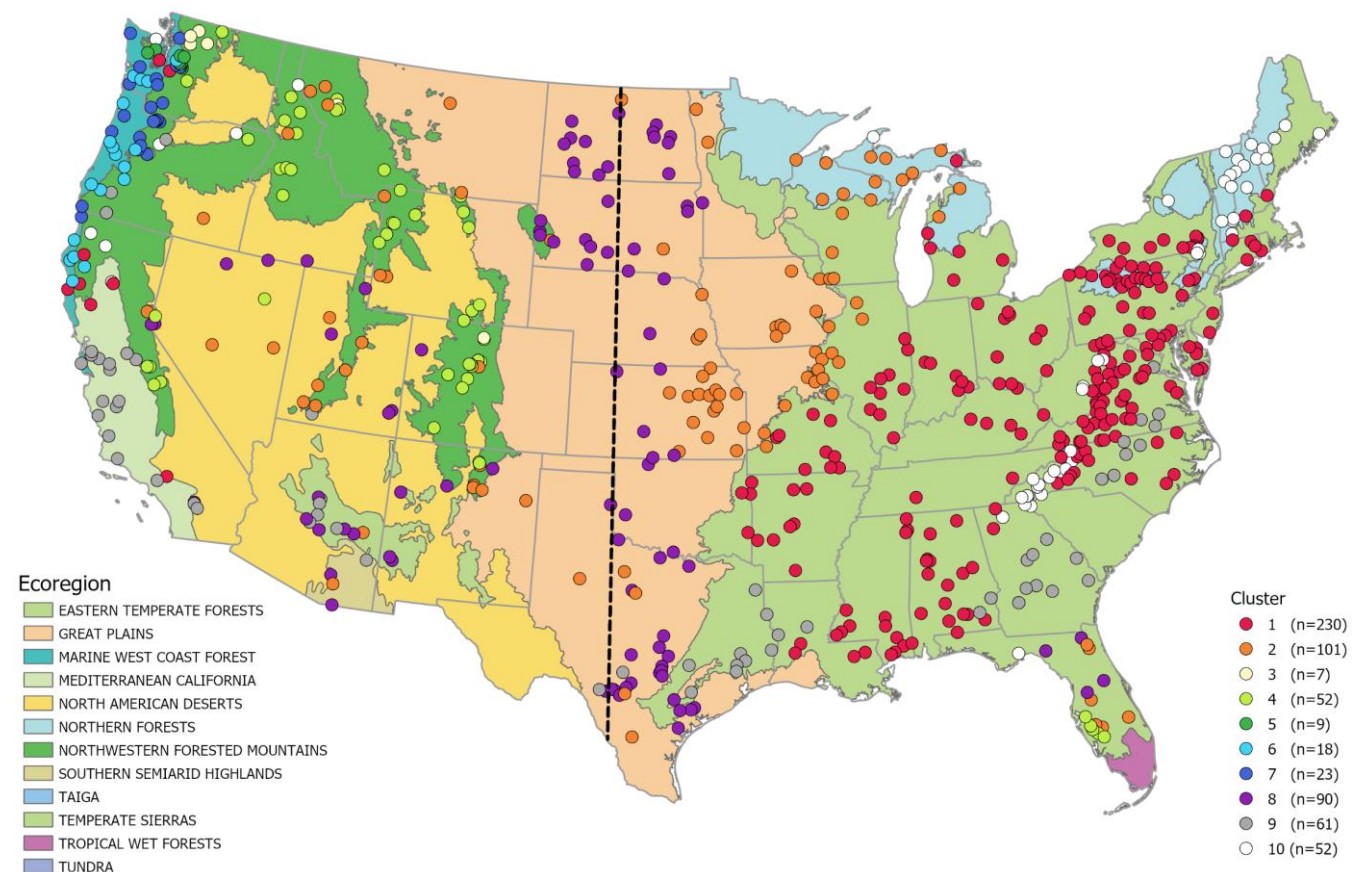


**Figure 4: Locations of the clustered CAMELS catchments and level I ecoregions (Omernik and Griffith, 2014) in the continental US. Dotted line marks the 100th meridian.**

We can see a split of the clusters along the 100th meridian. Cluster 3, 4, 5, 6 and 7 are located mainly in the west, while Cluster

1 and 10 are mainly found in the east. However, the remaining Clusters 2, 8 and 9 have roughly similar numbers of catchments in both regions. Overall, the catchments in the eastern half of the United States form large spatial patterns of similar behavior, while the catchments in the west are patchier. This same pattern can also be seen in some of the signatures used by Addor et al. (2018). Especially the runoff ratio and mean annual discharge form very similar patterns to the clusters in this study.

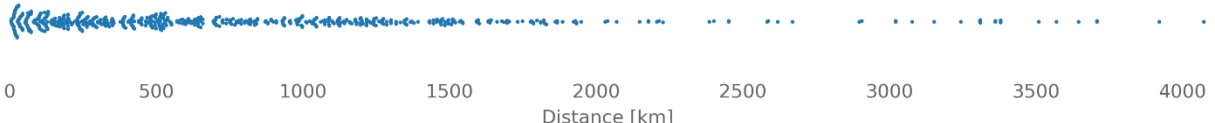

Figure 5: Swarm plot of the real world distances of all catchments to the most hydrologically similar catchment (based on their distance in the PCA space of the hydrological signatures).

In addition, similar catchments can be quite far away from each other (Figure 5). Sometimes, the catchment with the most similar signature was found as far as 4,000 km away (almost the entire longitudinal distance of the continental US). This explains why spatial proximity seems to be important in some studies that look into explanations of catchment behavior (Andréassian et al., 2012; Sawicz et al., 2011), but not in others (Trancoso et al., 2017). This also indicates that clustering by using spatial proximity might only work in regions like the eastern US, where the behavior of rivers changes only gradually, due to uniform climate that only changes gradually as well. The finding that the most similar catchment (based on their hydrological signatures) can be far away, also explains the behavior of clusters that contain catchments quite distant from each other (e.g. Cluster 4). Even though the catchments might be far away from each other, the interplay of different catchment attributes and driving factors, including sometimes very different climates, can lead to similar (equifinal) discharge behavior, concerning the overall amount of discharge, its distribution in the year, the high flows and the runoff-ratio. This was also found by several other studies (e.g. Berghuijs et al., 2014; Knoben et al., 2018; Kuentz et al., 2017).

In the following, we describe the catchment clusters in regard to their characteristics in meteorology (Figure 6), attributes (Figure 7), hydrology (Figure 8) and location (Figure 4). The main points of this description are summarized in Table 2. A list of all catchments with index, position, cluster classification and climate indices is given in the supplementary material.

**Cluster 1** is defined by a dense vegetation cover (Figure 7). The low elevation of those catchments results in little annual snow fall. They are mainly located in the southeastern and central plains and therefore get relative high rainfall (>1,000 mm per year) (Figure 4), almost uniformly distributed over the year (Figure 6). Still, they produce only little discharge. This cluster contains the highest number of catchments (n=230). So over one third of the catchments in CAMELS show a relatively similar behavior when it comes to the amount of water fluxes and their distribution throughout the year. This is particular visible when we look at annual supply of discharge (Figure 6). Even though the cluster contains a large number of catchments that also partly differ a lot in their potential evapotranspiration, there is only a minor difference in the amount of discharge and its seasonality.

**Cluster 2**'s most typical attribute is its high precipitation seasonality. However, concerning most other catchment attributes, Cluster 2 is undefined as it contains catchments of most regions of the continental US (with a concentration in the eastern Great Plains) (Figure 4). The hydrological signatures on the other hand show a clearer pattern. Here, the mean winter discharge, Q95 and the mean annual discharge have a narrow range (Figure 8). This shows that catchments with very different attributes can produce similar discharge characteristics. The different attributes seem to cancel each other out in their influence on the discharge. This might be enhanced by the high precipitation seasonality with higher precipitation in the summer, which creates a strong climatic forcing and thus a narrow range for the hydrological signatures (Figure 6). This cluster differs from the first one, by having even lower discharge, with almost no peaks and a higher influence of snow melt.

**Cluster 3** is the smallest cluster with only seven catchments. Those are all located in the Northwestern Forested Mountains. Their most distinct feature is their strong negative precipitation seasonality (indicating a strong precipitation peak in the winter) (Figure 6, 7). They also experience high precipitation events (mostly as snow). Hydrologically, their most distinct features is the very high mean summer discharge and high runoff ratio (Figure 8). This is probably caused by the large amounts of snow melt in late spring and early summer. The catchments of Cluster 3 have the largest snow storage in the dataset with mean maximum value of over 600 mm. Overall, the catchments in this cluster seem to be, from a hydrological point of view, the most extreme in the overall CAMELS data set. This can be seen in their varying discharge patterns. The uniting pattern is their large peak discharge during summer and their extreme values in the PCA space (indicating much higher values for the hydrological signatures in comparison with the other catchments) (Figure 3).

**Cluster 4** is, as cluster 3, located in the Northwestern Forested Mountains, with the exception of four catchments that are located in Florida (Figure 4). This cluster is another example of different catchment attributes being able to create similar discharge characteristics concerning the signatures used, while having very different catchment attributes (Figure 6). The catchments have overall low discharge and few high flow events, except one large peak in the mid of the summer, which is caused by melting snow in the northern catchments and strong rainfalls in Florida. Their catchment attributes vary widely, especially in all attributes that are related to elevation (e.g. fraction of precipitation falling as snow) (Figure 7), which is to be expected when some of the catchments are located close to the sea in the southeast, while others are mountainous.

**Cluster 5** includes only few catchments (n=9), which are all located at regions in the northern part of the Marine West Coast Forests (Figure 4). This is the region in the continental US that receives the highest precipitation (>2000 mm year), which is reflected in their discharge characteristics (Figure 6, 8). These catchments have the highest discharge in the whole dataset, especially in the early summer, due to a combination of high precipitation and snowmelt. They also experience only few high precipitation events as they receive large amounts of rain and snow most of the year, with a distinct very high peak in the

winter months. They further depict an additional discharge peak in late spring/early summer that separates them from the other

280 catchments found at the west coast. The catchments are uniformly covered by almost 100% of forest.

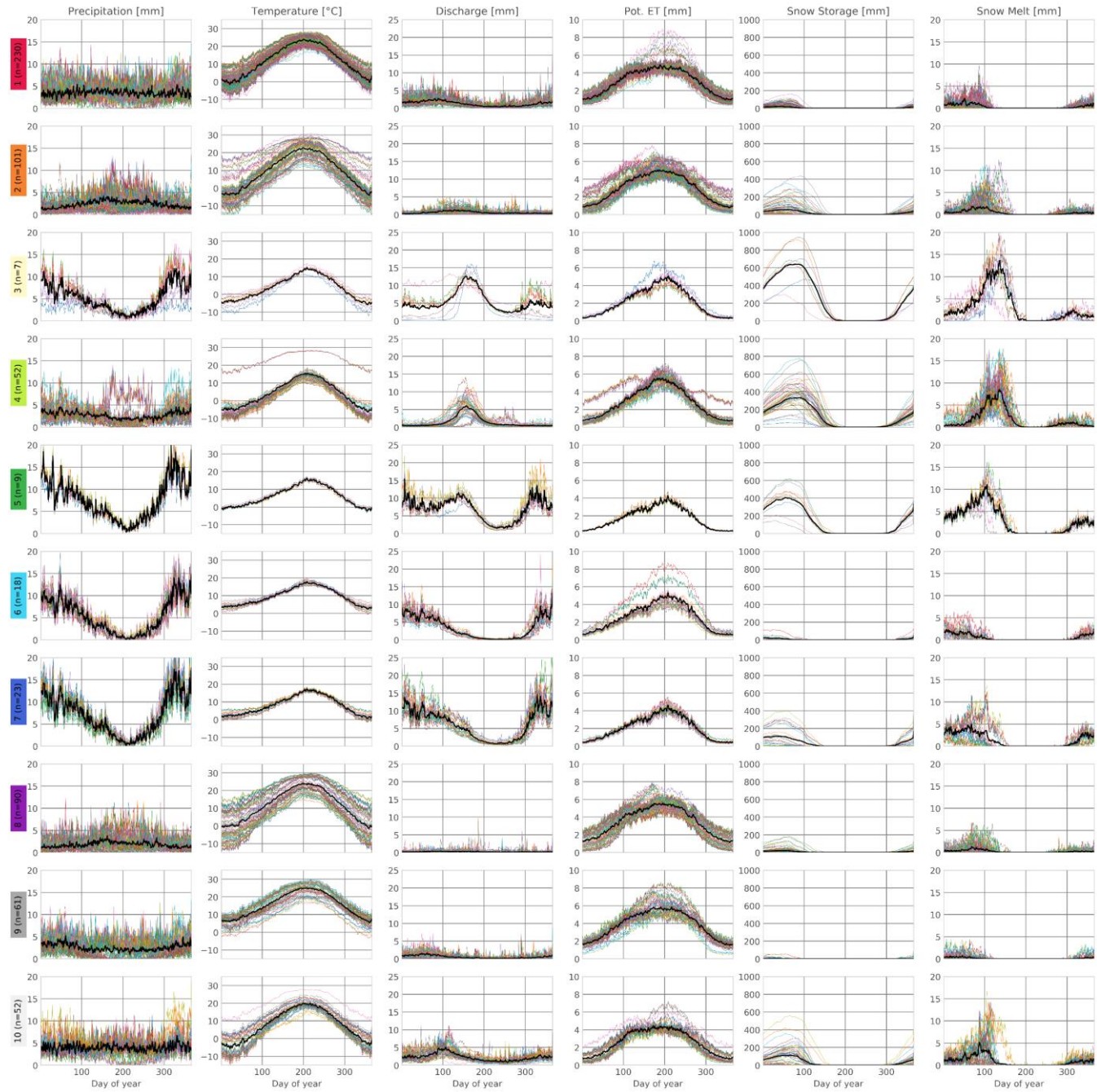

**Figure 6: Meteorological attributes of the clustered CAMELS catchments averaged by day of the year. Potential Evapotranspiration (Pot. ET) was calculated with Hargreaves-Samani (Samani, 2000). Snow storage and melting was calculated using a temperature based approach described in Massmann (2019). Black lines indicate the mean of all cluster members. Colored lines represent the individual catchments.**

**Cluster 6** is located in the Marine West Coast Forest, but in contrast to Cluster 5, it covers the whole region and not only the northern part (Figure 4). The catchments are very similar in their attributes and discharge characteristics to Cluster 5, with the exception of lower discharges and runoff ratios (Figure 7, 8). This is caused by slightly lower precipitation in comparison with Cluster 5. Cluster 6 experiences the most negative precipitation seasonality across all clusters, with almost all precipitation falling in the winter month. Due to this seasonality and the lower precipitation in the summer, the catchments of this cluster uniformly dry out almost completely in the late summer (Figure 6).

**Cluster 7** is also located in the same region as Cluster 5 and 6 (Marine West Coast Forests) (Figure 4). Concerning the catchment attributes and the discharge characteristics, it is located between Cluster 5 and 6. So, Cluster 5 to 7 all cover the same region and differ in their mean summer discharge, which is caused by variations in elevation and location (Figure 7). Cluster 7 has higher subsurface permeabilities than cluster 6, which might explain the differences in hydrological behavior, even though the overall attributes of both clusters are rather similar. For example, Cluster 7 has an overall lower discharge than Cluster 5, but does not dry out during the summer as Cluster 6 does (Figure 6). This might be due to the larger amount of snow it receives in comparison with Cluster 6 and its lower evapotranspiration.

**Cluster 8** is the most arid cluster (Figure 7). All of the catchments are located in western parts of the Great Plains and in the North American Deserts (Figure 4). They are characterized by an overall low water availability and high evaporation, which is shown in the very low mean annual discharge and runoff ratio (Figure 6, 8). This also results in low values for the LAI. Yet, the frequency of high precipitation events is high. However, those high precipitation events are only high in comparison with the mean precipitation for those catchments and not the overall range of precipitation in the entire CAMELS dataset.

**Cluster 9** covers all southern states of the United States (Figure 4). The catchments here are quite similar to Cluster 8, but show a lower precipitation seasonality and a higher forest cover and green vegetation (Figure 7). In addition, all catchments of this cluster are in relative close proximity to the sea. The uniting factor in this cluster seems to be the very low snow fraction and the high evapotranspiration (Figure 6, 7).

**Cluster 10** catchments are all located in the Appalachian Mountains (Figure 4). The mean elevation is higher than of most other clusters and the catchments have a low aridity and a very high forest cover (Figure 7). Their discharge characteristics are similar to that of the Marine West Coast Forests Clusters 5 to 7 (Figure 6, 8). However, they receive less water than those catchments. Cluster 10 covers the same ecoregion as Cluster 1, but has a distinct behavior due to its mountainous character, which can be seen in the higher seasonality of the discharge. This is probably caused by the larger snow cover, with a discharge peak in spring due to snow melt.

Overall, we can see similar trends for some of the cluster. We identified four distinct groups. The general similarities of the clusters are also represented by their distance and position in the PCA space (Figure 3).

- Group 1 (Cluster 1, 2, 8, 9): low seasonality in precipitation and discharge; located in the eastern US; due to low slope inclinations, water takes a long time to reach the outlet.
- Group 2 (Cluster 3, 4): dominant summer peak of discharge caused by rapid snow melt; mostly located in the mountains of the western US; differ in precipitation inputs.
- Group 3 (Cluster 5, 6, 7): located in the Northwestern Forested Mountains; characterized by high precipitation amount and seasonality, but more or less extreme versions.
- Group 4 (Cluster 10): located in the Appalachian mountains; share characteristics with Group 1, though influenced by higher elevations and steeper slopes.

Those groups of clusters are similar to the ones found by (Berghuijs et al., 2014), even though they used a very different method to derive them. The main difference in the groups is probably caused by how we structure the clusters and groups in the eastern US, due our clusters being more influenced by the Appalachian Mountains. However, both approaches deliver similar results overall.

The question remains: what is the right numbers of clusters? Though even we did find four distinct groups, having only four clusters would probably be too little, as the clusters in the groups show a wide range of behaviors (Figure 3, 7, 8, Table 2). There are catchment attributes, which we did not take into account, but which could further split up the clusters (e.g. the shape of the catchments). However, this study considered the catchment attributes that are usually considered as being important. The fact that the clusters contain different numbers of catchments can be explained by their distances in the PCA space (Figure 3). Many of the catchments are rather similar. This produces some clusters which contain most of the catchments. However, we also have some extreme catchments (e.g. Cluster 3 and 5), which are very different to the bulk of the catchments in the CAMELS dataset. Thus, even though some of our presented clusters are quiet small in number, they are needed to capture their extreme hydrological behavior. It can also be seen that for most of the clusters there is no clear dividing line to neighboring clusters. Therefore, it might be useful to use fuzzy clustering approaches in future research, to avoid those strict boundaries in a continuous space. Our results show that some of the clusters follow the boundaries of the ecoregions in the US very directly (Cluster 1), while others do not (Cluster 9). The worlds of ecology and hydrology are sometimes shaped by the same forcing, but not always.

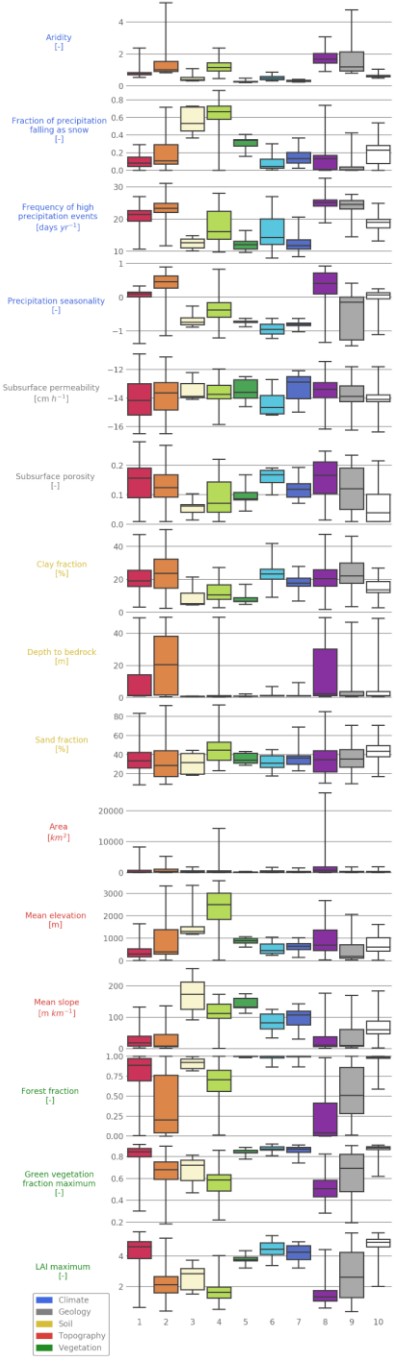

**Figure 7: Boxplots of the catchment attributes of the clusters**

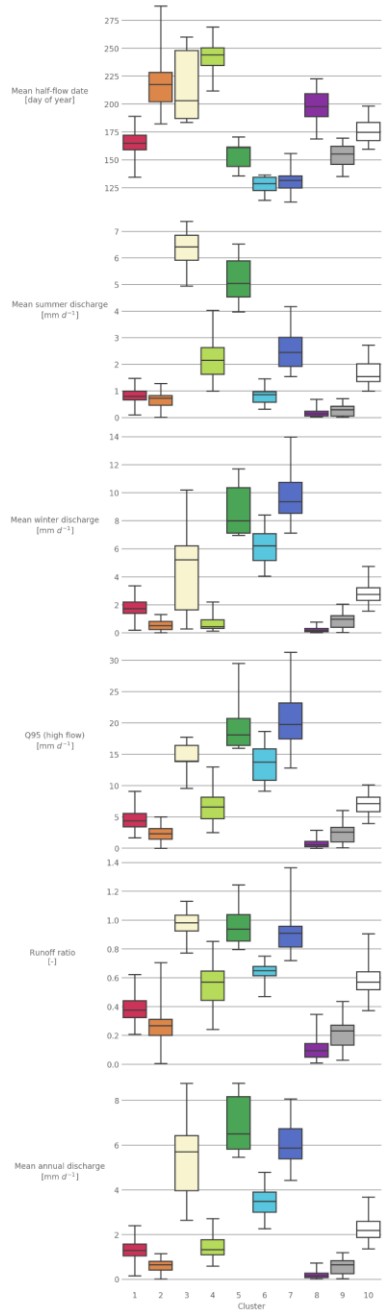

**Figure 8: Boxplots of the hydrological signatures of the clusters.**

 **Table 2: Properties of the catchment clusters. Typical signatures/attributes refers to the signature/attribute of the cluster with the lower coefficient of variation scaled by the mean coefficient of variation of the whole dataset. Dominating attribute refers to the catchment attribute that has the highest weighted $R^2$.**

| Cluster | n | Main Region | Typical signature | Typical attribute and their manifestation | Dominating attribute |
|---|---|---|---|---|---|
| 1 | 230 | Southeastern and Central Plains | Low mean winter discharge | Low aridity | Aridity |
| 2 | 101 | Central Plains (with scattered catchments all over western US) | High mean half-flow date | High precipitation seasonality | Green vegetation fraction maximum |
| 3 | 7 | Northwestern Forested Mountains | High mean summer discharge | Low precipitation seasonality | Fraction of precipitation falling as snow |
| 4 | 52 | Northwestern Forested Mountains and Florida | High mean half-flow date | Mid frequency of high precipitation events | Precipitation seasonality |
| 5 | 9 | Northern Marine West Coast Forests | High mean summer discharge | Very high forest fraction | Forest fraction |
| 6 | 18 | Marine West Coast Forests | Mid runoff ratio | Low precipitation seasonality | Aridity |
| 7 | 23 | Western Cordillera (Part of Marine West Coast Forests) | High mean winter discharge | Low precipitation seasonality | Fraction of precipitation falling as snow |
| 8 | 90 | Great Plains and North American Deserts | Mid mean half-flow date | High frequency of high precipitation events | Precipitation Seasonality |
| 9 | 61 | All southernmost states of the US | Low mean half-flow date | High frequency of high precipitation events | Aridity |
| 10 | 52 | Appalachian Mountains | Low mean winter discharge | High forest fraction | Mean elevation |

**3.5 Importance of the catchment attributes in the clusters**

The individual importance of the catchment attributes in the clusters is variable and partly deviates from the order of importance in the overall dataset (compare Figure 2 and Figure 9). For Cluster 1 (Southeastern and Central Plains), 6 (Marine West Coast Forests) and 9 (coastal states) aridity has the highest weighted coefficient of determination in the clusters. For Cluster 3 (Northwestern Forested Mountains) and 7 (Western Cordillera) the highest relevance is found for the fraction of precipitation falling as snow. For the remaining clusters it is precipitation seasonality (Cluster 4 (Northwestern Forested Mountains), Cluster

8 (Great Plains and Deserts)), the green vegetation fraction maximum (Cluster 2 (Central Plains)) and the mean elevation (Cluster 10 (Appalachian Mountains)). We can also see that some clusters have one dominating catchment attribute (investigated by the coefficient of determination e.g. aridity in Cluster 1, compare Figure 9), while for other clusters, all attributes seem equally important (e.g. Cluster 8). Overall, the western clusters (west of the 100th meridian) display the highest weighted $R^2$ with:Fraction of precipitation falling as snow (Cluster 3, 7)

-      Precipitation seasonality (Cluster 4)

         -      Forest fraction (Cluster 5)

         -      Aridity (Cluster 6)

eastern clusters (east of the 100th meridian) with:

         -      Aridity (Cluster 1)

-      Mean elevation (Cluster 10)

clusters equally present in west and east with:

         -      Green vegetation fraction maximum (Cluster 2)

         -      Aridity (Cluster 9)

         -      Precipitation seasonality (Cluster 8)

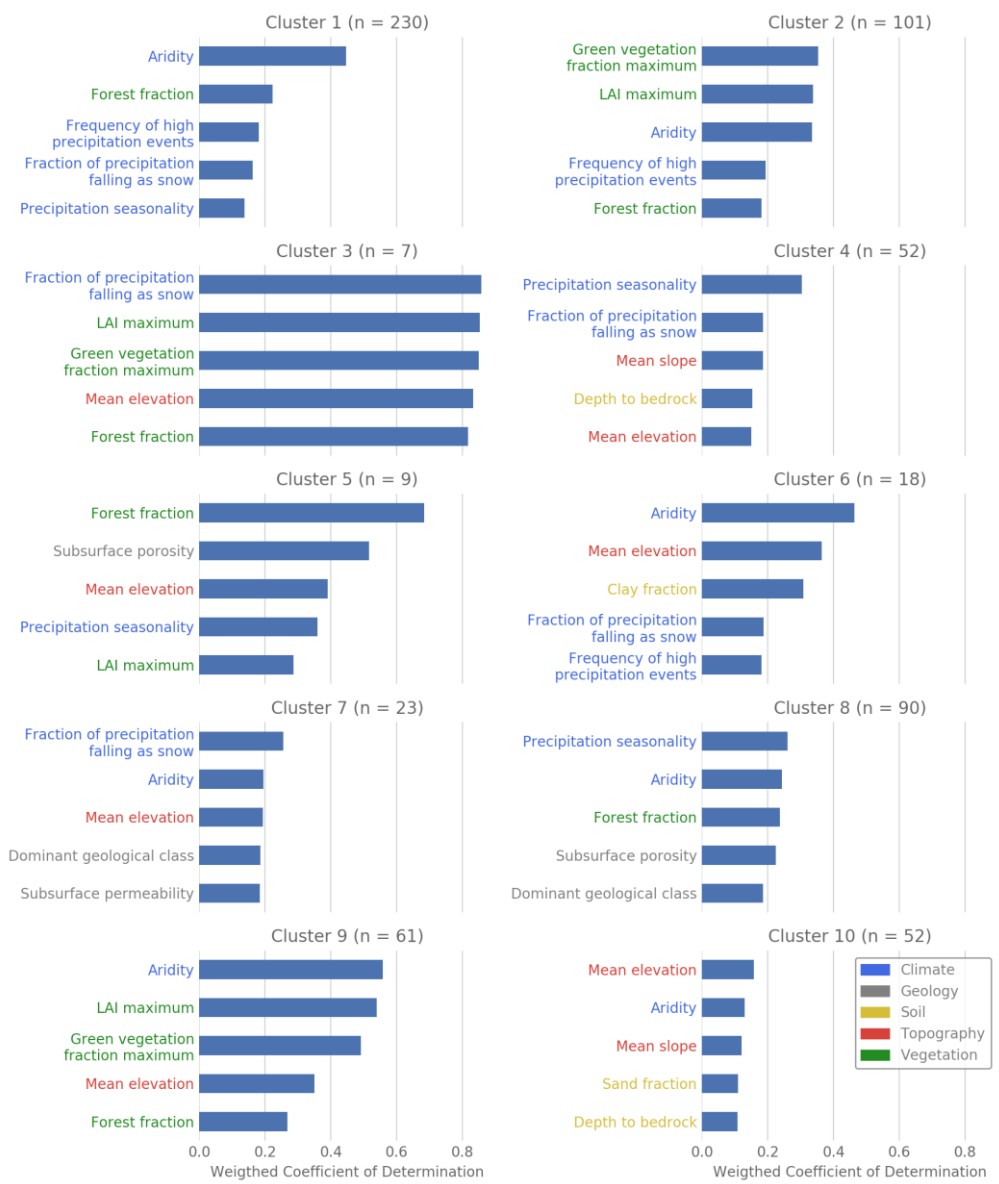

**Figure 9: Importance of the catchment attributes evaluated by the quadratic regression for the catchment clusters. Attributes colored according to their catchment attribute class.**

Keeping the correlation coefficients displayed in Figure 1 in mind, we see that climate is the most important factor in almost

all clusters, as the vegetation attributes are highly correlated with the climate attributes. The only exception is Cluster 10 in

which mean elevation is the most important catchment attribute. However, the catchment attributes in Cluster 10 have overall

low R² values and the mean elevation is directly followed by the aridity. This again shows that climate seems to be the dominating factor for catchment behavior, as found in other large sample studies (e.g. Berghuijs et al., 2014; Kuentz et al., 2017). Nevertheless, if one takes a closer look at the data set, more detailed, regional correlations with regard to individual climate variables can be determined. For example, Cluster 1 is defined by the aridity, while Cluster 4 seems to be much more influenced by the precipitation seasonality. Overall, it is feasible to link dominating catchment attributes to the hydrological behavior. While it is straightforward in some regions of the US, it is more challenging in others. We link this to the signal of the climatic forcing being more superimposed by other catchment attributes, which results in a less clear connection between its hydrological behavior and the climate. This hints that climate and catchment attributes are more intertwined in those areas and indicates regions where different types of hydrological runoff generation processes are existing. Furthermore, it indicates regions where hydrological predictions in ungauged basins (Hrachowitz et al., 2013) can become very challenging, as the interplay of the available meteorological data and catchment-attributes cannot sufficiently explain the hydrological characteristics. Those findings also highlight one current discrepancy between large sample and single catchment studies. While large sample studies, especially the very large ones, identify climate as being most important for the hydrological behavior (e.g. Addor et al., 2018; Kuentz et al., 2017), smaller sample studies (e.g. Chiverton et al., 2015; Pfister et al., 2017) and single catchment studies (e.g. Floriancic et al., 2018) often identify the geology or soils as being very important. This might be linked to the overall problem of scales in hydrology, as different scales of soil/geology and climate have different effects and varying data accuracy (Addor et al., 2018; Blöschl, 2001). In addition to this, the overall scale might also come into play. Smaller sample studies often compare catchments that are not far away from each other and probably have similar climate forcings. Thus, the differences in hydrological behavior can only be caused by catchment attributes other than climate. Therefore, larger and smaller sample studies might be looking at different things. While very large sample studies capture what drives catchments on large scales (the climate), smaller studies look at how this climatic signal is transferred to discharge by the catchment attributes.

## 3.6 Differences in clusters in comparison with other hydrological clustering studies

The results of this study show some similarities with the clustering results of Kuentz et al. (2017), who derived their cluster from European catchments by an analogous method. Like them, this study here also found one cluster (Cluster 2) that does not have any distinct character. However, only around one sixth of the CAMELS catchments belongs to this Cluster 2, while Kuentz et al. (2017) had one third of their catchments in a cluster without distinct features. Therefore, our selection of hydrological signatures seems to allow a better identification of hydrological similarities. However, all catchments in CAMELS are mostly without human impact (Addor et al., 2017), while many catchments in the study of Kuentz et al. (2017) are under human influence. This human influence might mask otherwise apparent patterns. Kuentz et al. (2017) also found

two clusters that contain mostly mountainous catchments. These show a similar behavior to Cluster 3 (Northwestern Forested Mountains) and Cluster 10 (Appalachian Mountains) (Figure 4). The main difference between their findings and this study here is Cluster 8, as it contains very arid catchments (with some being located in deserts). Obviously, this cluster cannot be found in Europe as Europe has no real deserts. Still, there is some similarity with their cluster of Mediterranean catchments as both are dominated by aridity. Summarizing, in their study and this study catchments are mainly clustered in groups of desert/arid catchments, mountainous catchments, mid height mountains with high forest fraction, wet lowland catchments and one cluster of catchments that do not show a very distinct behavior and therefore do not fit in the other clusters (Table 2). One possible explanation for this unspecific behavior might that many catchments have one or two important attributes that dictate most of their behavior, but which are different from other cluster members. For example, desert catchments are relatively easy to identify, as they are dominated by high energy and little precipitation. A European upland catchment on the other hand has several more influences such as snow in the winter, high energy in the summer, varying land use and strong impact of seasonality. Here, many influences overlap each other and make it thus difficult to identify a single causes, see also the discussion by Trancoso et al. (2017) that goes in a similar direction. Those overlapping influences are probably also the reason why catchment classification studies often find clusters where one or two clusters that include a large number of catchments, while most other cluster only contain few catchments (Coopersmith et al., 2012; Kuentz et al., 2017). Therefore, it is quite difficult to confirm the 'wish' of the hydrological community to have homogenous catchment groups with only a few outliers (e.g. Burn 1997), because catchments are complex systems with a high level of self–organization arising from co-evolution of climate and landscape properties, including vegetation (Coopersmith et al., 2012). Accordingly, it requires many separate clusters to separate those multi-influence catchments into homogenous groups. This hints that for future research a fuzzy clustering approaches might provide less ambiguous results, as it respects the continuous nature of hydrological behavior. Still, the cluster found here might capture much of the variety present in the United States, as they roughly follow ecological regions (McMahon et al., 2001), which has been stated as a hint of a good classification (Berghuijs et al., 2014). In addition, this study shows that using clusters derived from principal components of hydrological signatures creates meaningful groups of catchments with similar attributes (Figure 6, 7, 8). Those clusters also show distinct spatial patterns (Figure 4). Similar results were also found in other studies that used the same method (Kuentz et al., 2017; McManamay et al., 2014), but based them on partly different hydrological signatures. Therefore, the principal components of hydrological signatures can be used as a measure of similarity between catchments. They represent the "essence" of all hydrological signatures used. Our results also show that it is difficult to link those catchment clusters to simple averaged measures of catchment attributes. While some clusters have very clear connections to the attributes, others have no catchment attribute that could easily explain the behavior of the catchments. This hints, that some catchments are easier to explain (in a hydrological sense) than others. Those difficulties might be an artifact of the averaged catchment attributes or be caused by complex catchment reaction, forced by intertwined climate and catchment attributes. Which in turn, might indicate an equifinality of catchment response.

**3.7 Comparing catchment clusters based on hydrological behavior and climate**

Besides hydrological behavior, climate is often used to sort catchments into similar groups (e.g. Berghuijs et al., 2014; Knoben et al., 2018). Therefore, we are interested if both approaches deliver comparable results. To evaluate this, we contrasted our results to the commonly used Koeppen-Geiger climate classification (Beck et al., 2018) (Figure 10) and recently published approach of Knoben et al. (2018), who sorted climate along three continuous axis of aridity, seasonality and fraction of precipitation falling as snow (Figure 11). The resulting clusters based on climate and hydrology should be the same, if climate is the dominating driver of hydrological behavior in every catchment. Yet, this is not the case for the Koeppen-Geiger classification. In every hydrological cluster are at least two different climates regarding the Koeppen-Geiger classification, ranging up to eight different climatic regions for Cluster 2 and 8 (those even include deserts and very cold regions). Thus, the Koeppen-Geiger classification seems unable to capture the essential drivers of hydrological behavior. A critique also raised in other studies (e.g. Haines et al., 1988; Knoben et al., 2018).

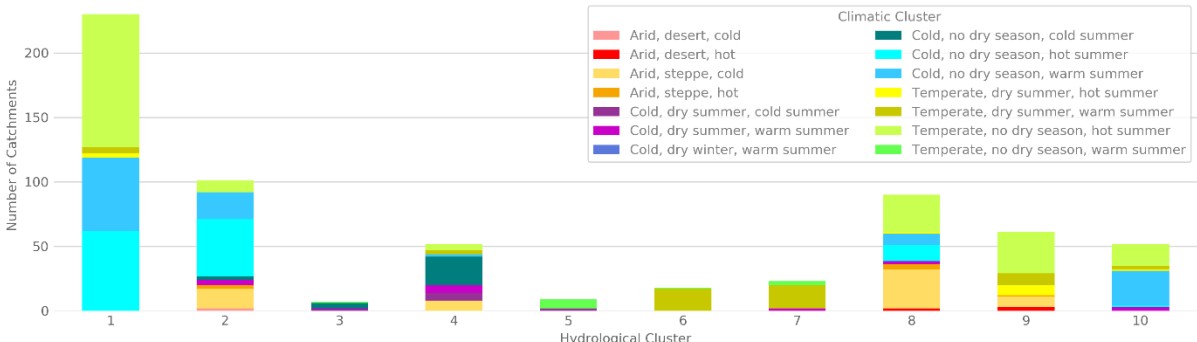

**Figure 10: Membership of Koeppen-Geiger clusters (Beck et al. (2018)) in the hydrological clusters.**

The picture is less clear concerning the climatic index space of Knoben et al. (2018) (Figure 11a). Due to the continuous nature of the approach of Knoben et al. (2018), there are no clear boundaries as in the Koeppen-Geiger classification. Still, there are some emerging patterns. For example, according to the approach of Knoben et al. (2018) Cluster 1 is mainly defined by a relatively arid climate, with some seasonal variability and little to no snow. This is in line with our analysis of the most influential catchment attributes for this cluster, as we identified aridity as the main driver. There seem to be regions were the forcing signal of the climate is transferred more directly to a streamflow response than in others. However, this does not mean that climate is unimportant in those regions. Either the climate forcing signal is changed more through other attributes of the catchment, or the mean values describing the climate do not properly reflect the variability of the climate in the single catchments. This leads to less clear correlation between the climate and the hydrological behavior. Interestingly, when we look at the single hydrological signatures in the climate index space (Figure 11b, A2) we see a very clear connection between the single hydrological signatures and the climate. This direct connection of the signatures used was also found by Addor et al.

(2018). Our results and the comparison show that the complex hydrological behavior, captured in a range of hydrological signatures, does not simply follow the climate only, even though the individual signatures do. Still, all signatures combined seem to capture a dynamic, which is climatic in origin, but is shaped through the attributes of the catchments (like vegetation and soils (Berghuijs et al., 2014)). Therefore, to find truly similar catchments, using climate characteristics only, is probably not sufficient (see also Addor et al., 2018; Knoben et al., 2018; Kuentz et al., 2017).

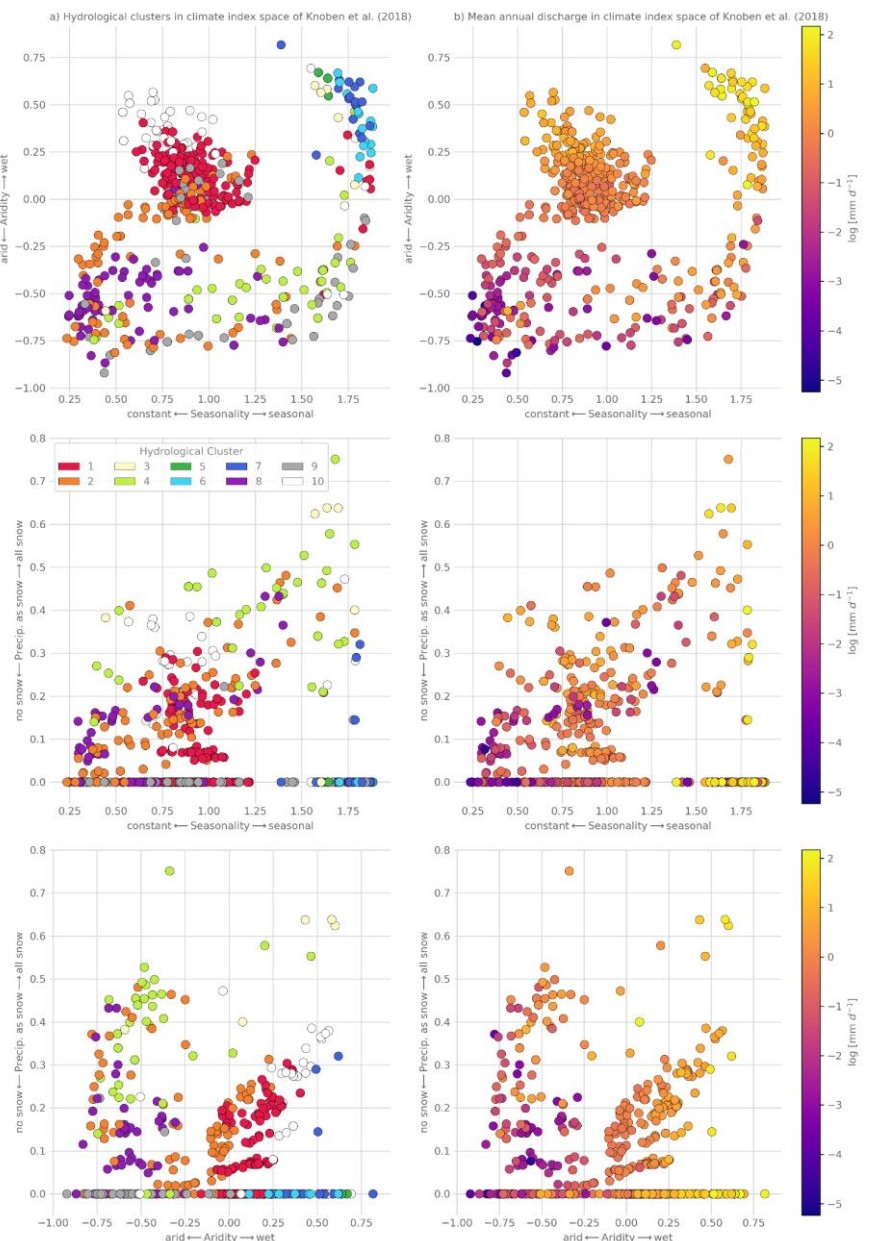


**Figure 11: a) Comparison of the hydrological clustering of this study with the climate index space of Knoben et al. (2018). Single dots show the catchments and are colored by their hydrological clusters. b) Mean annual discharge for all catchments in the climate index space of Knoben et al. (2018). Single dots show the catchments and are colored according to the value of the mean annual discharge. The log of the mean annual discharge is used to show the relative differences between the catchments. For a depiction of**
**all hydrological signatures used, see Figure A2.**

**4 Summary and conclusion**

This study explored differences in the catchment characteristics between the eastern and western US, the properties and location of catchment clusters based on hydrological signatures, the importance of catchment attributes for those clusters and how this study relates to other clustering studies and methods. We found that the correlations catchment characteristics are

quite similar for the eastern and western US with the exception of mean elevation, snow, geology and the leaf area index. For the overall CAMELS data set climate seems to be the most important factor for the hydrological behavior. However, depending on the location either aridity, snow or seasonality were most important. The clusters derived from the hydrological signatures partly follow the eco regions in the US and can combined into four groups of general behavior trends. Still, similar catchments can be quite far away from each other. We also found that most of the catchments have a rather similar discharge behavior,

while only some, more extreme catchments, derivate from that main trend. This might be a hint why it is so difficult to clusters catchments, as those single extreme catchments are quite unique and do not fit together well with other catchments. We also found, that there are differences of how directly the signal of forcing climate can be found again in the hydrological behavior. This explains why catchments often show a surprisingly similar behavior across many different climate and landscape properties (Troch et al., 2013) and why the most hydrologically similar catchment can be hundreds of kilometers away. Those

findings also relate to the paradox that small scales/single catchment studies identify geology/soils as most important for the hydrological behavior, while large sample studies usually find the climate as most important. This might simply be influenced by spatial proximity. Small scale studies look at catchment which all have a similar climatic forcing and thus only the other catchment attributes can be the cause of differences in hydrological behavior. Large sample studies on the other hand consider catchment from a wider area and thus attribute the differences in behavior to climate.

The aggregated data used in this study might level out the variability of the catchment attributes in the single catchment, but it also indicates that there is a kind of equifinality in the behavior of catchments. Different sets of intertwined climate forcing and catchment attributes could lead to a very similar overall behavior, not unlike to hydrological models that produce the same discharge with different sets of parameters.

We acknowledge that the results are dependent on the amount and size of the clusters, the catchment attributes considered and

the hydrological signatures used. Still, we think that the CAMELS dataset offers an excellent overview of different kinds of catchments in contrasting climatic and topographic regions. In addition, this study shows that using hydrological signatures with high spatial predictability results in hydrological meaningful clusters, which show consistent low flow behavior, even though those low flows were not explicitly considered. However, it seems that even a comprehensive dataset like CAMELS, does not allow an easy way to find a conclusive set of clusters for catchments. For future research, we recommend to include

measures of spatial variability of the climate in the single catchments and to look into the single clusters in more depth. This

might help to prove, if a less clear climatic signal is caused by intra-catchment variability of the climate or a larger influence of other catchment attributes.

**Data availability**

The CAMELS dataset can be found at https://ncar.github.io/hydrology/datasets/CAMELS_timeseries and is described in
Addor et al. (2017). The cluster numbers together with the CAMELS catchment ID and the climatic indices can be found in the supplement of this paper.

**Code availability**

The code used for this study can be found at Jehn (2018).

**Author contribution**

FUJ, LB, TH and PK conceived and designed the study. FUJ did the data analysis. All authors aided in the interpretation and discussion of the results and the writing of the manuscript.

**Competing interests**

The authors declare that they have no conflict of interest.

**Acknowledgment**

We would like to thank Ina Pohle, Marc Vis, Jan Seibert, Wouter Knoben, Andrew Newman and two anonymous reviewers for giving valuable and important feedback in the creation of this paper. We would also like to thank all the people who helped create the CAMELS dataset. Thank you for your work! We further would like to thank the DFG for generously funding the project HO 6420/1-1.

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

**Appendix**

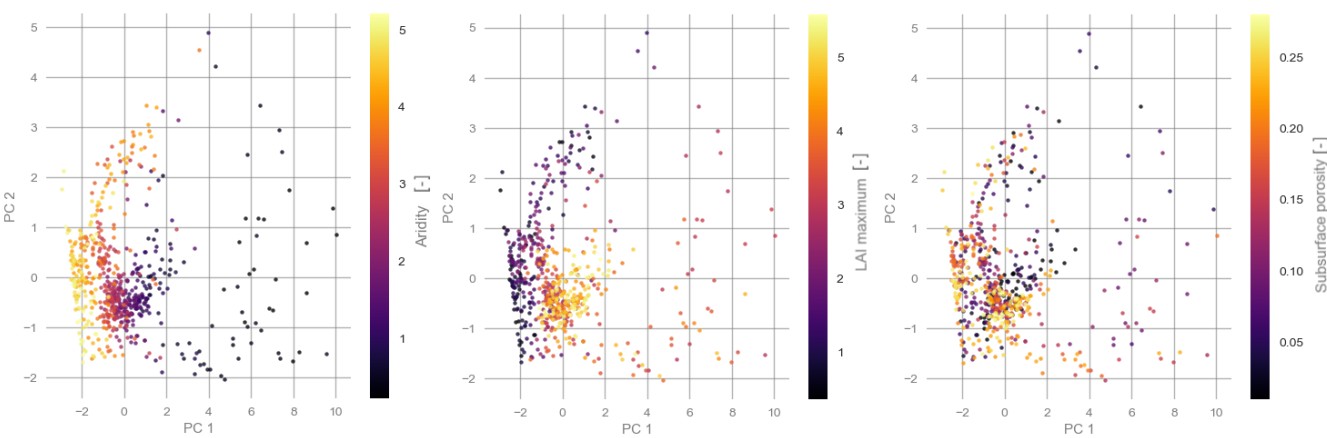

**Figure A1: Patterns of catchment attributes in the PCA space of the hydrological signatures, with decreasing strength of the observed pattern from left (aridity) to right (subsurface porosity).**


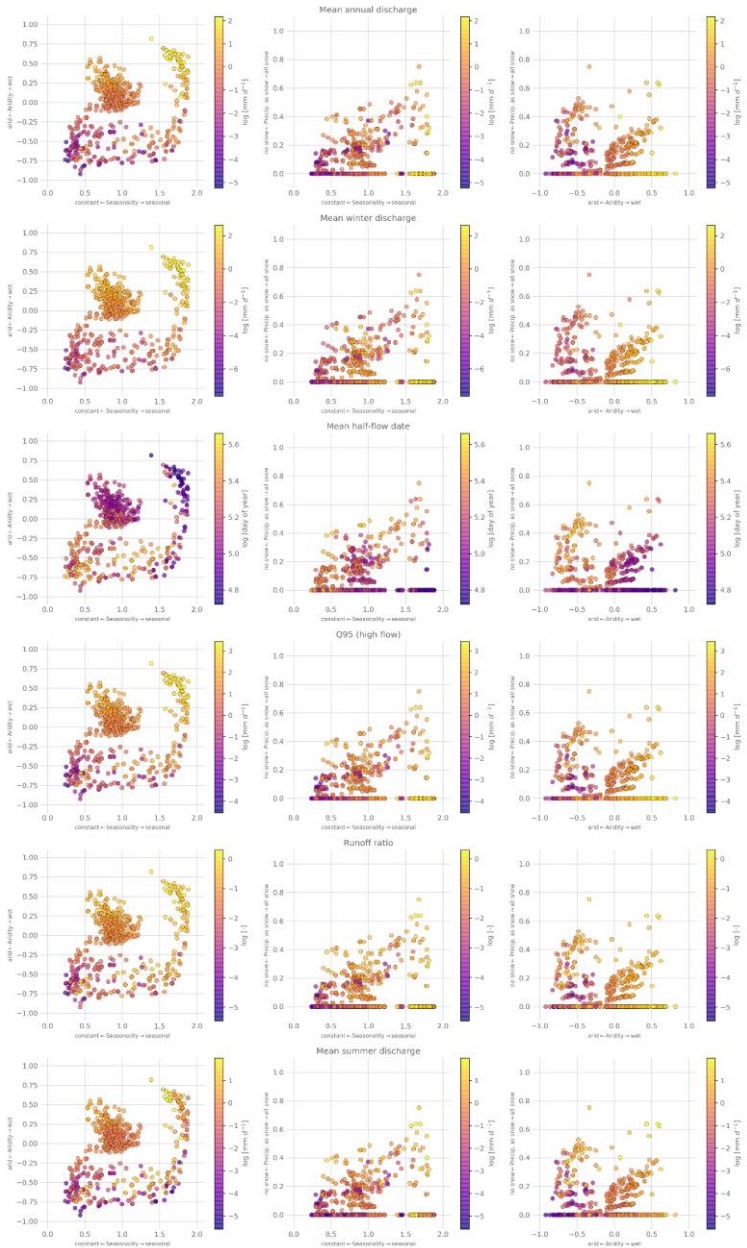

**Figure A2: Hydrological signatures for all catchments in the climate index space of Knoben et al. (2018). Single dots show the catchments and are colored according to the value of the mean annual discharge. The log of the signatures is used to show the relative differences between the catchments.**