# Peer review of "Using hydrological and climatic catchment clusters to explore drivers of catchment behavior."

_Hydrology and Earth System Sciences, 2019_

## Referee Comment (RC1) · Anonymous Referee #1 · 29 Apr 2019

Jehn et al. classified the CAMELS catchments based on hydrological signatures, and subsequently investigated the link between catchment attributes and the classes. The conclusion of the study is that catchment behavior can mainly be attributed to climate in regions with homogeneous topography, but that this is more difficult in regions with heterogeneous topography.

Unfortunately, my perception is that the conclusions of the study are based on a fallacy. The main problem can be found here: "If climate were the main driver, the clusters would be located along a climatic gradient. However, this is only true for the eastern half of the United States (for a climatic map of the United states see (Beck et al., 2018). In this part of the United States, the low relief allows large regions with a uniform climate, that only changes of larger scales." If looking at the map in Beck et al.

[Figure]

(2018), but also Peel et al. (HESS, 2007), or Knoben et al. (WRR, 2018), indeed the eastern part of the US shows large regions with uniform climate. But the maps also all show the large scattering in climates in the west: there is more spatial variation in climate in the western part of the US. This therefore seems no justification to state that climate is less relevant in regions with varying topography - there, the climate is just more variable too. This is also confirmed by the results of the study, where precipitation falling as snow is found as one of the main indicators in the west. "This implies that climate is a good indicator for the discharge characteristics as long as the topography is homogenous." seems therefore a too strict and incorrect conclusion, that does not necessarily follows from the results / figures. Furthermore, I wonder to what extent 'homogeneous topography' can be found as criterion, when looking at the catchment scale, considering that most catchments in CAMELS are rather small.

Besides my disagreement with the main conclusion, I consider the insights gained from the paper low compared to already available literature, especially considering Addor et al. (2018) and Knoben et al. (WRR, 2018). What did we learn from this study about the relation between attributes and signatures, or catchment clustering, that was unknown before? Especially given that I disagree with the main conclusion. If is it the method applied (PCA combined with clustering), then further elaborate on the methods and better explain everything that is done and how it differs from other studies. This also needs explanation why this method would provide insights that cannot / haven't been obtained with other methods. I would like to encourage the authors to dive deeper into the material and expand the analysis, and have a critical look at their own conclusions.

---

## Short Comment (SC1) · 2 May 2019

Dear Referee 1,
Thank you for your comment, which point out two rather critical aspects. As you correctly pointed out in your first statement, climate is obviously an important factor in the western United States. We fully agree with this and it may well be that we have formulated this too hard in our manuscript. Still, we would like to point out that our conclusion is grounded in the observations. While in the overall dataset climate attributes are clearly the most important (Fig 2.), the case is different for the single clusters. Here, several non-climatic attributes are most important (Fig. 4, e.g. cluster 2). In cases where climatic attributes are most important, they are often only slightly more important than other attributes (e.g. cluster 4). We see this as a clear indication that

the climate in the western United States is not the dominating factor for catchment classification. However, we acknowledge that our wording does not depict this accurately enough and we will revise our manuscript accordingly.

The second point of criticism is that our analysis does not provide any new findings compared to other existing studies. However, the studies cited by Addor et al. (2018) and Knoben et al. (2018) have a different focus and followed a different methodological approach. Knoben et al. (2018) clustered their catchments using climate and linked this to a specific river behaviour. Addor et al. (2018) also starts from catchment attributes and related these with hydrological signatures. Our approach takes another scientific angle. We identified rivers with similar hydrological behaviour according to hydrological signatures and combined these with climatic information. This allows to show that relying only on climatic attributes in clustering catchments can be misleading, as we found catchments that belonged to a single cluster regarding their hydrological behaviour, but did have very different catchment attributes (e.g. cluster 4 with catchments in Florida and the Northwestern Forested Mountains). In addition, our study enables further research that looks into comparing rivers with similar behaviour. This will also be made more clear in a revised version of our manuscript.

Kind regards,

Florian Ulrich Jehn

---

## Referee Comment (RC2) · Andrew Newman (Referee) · 4 Jun 2019

General comments: This paper examines the CAMELS catchments and clusters them using hydrologic signatures that have been previously found to have high spatial predictability. Overall this study is somewhat unsatisfying. Little new physical insight is gained in understanding how we determine similarity across catchments. The results do agree with past studies, which is a good test of the previous work. However, what does this specific study bring to us? Previous results discussed here, e.g. Addor et al. (2018) (Fig. 4), and Newman et al. (2015) (Fig. 12) found the same results. Aridity is the primary driver of basin behavior given the catchment scale attributes used, followed by other climate indices (e.g. snow). Finally, one of the primary conclusions drawn from the clustering results needs to be reexamined (specific comment #2).

[Figure]

Specific comments:

1) Is 10 clusters necessary? Why does 10 make this study similar to others? Did those studies arbitrarily pick 10 also? I wonder if a similar cluster selection method is better, rather than the same number of clusters. A more detailed justification in the methods section is necessary.

2) Many of the attributes have high co-variability. For example, elevation and temperature/fraction of snowfall, elevation and mean slope, forest fraction and elevation (in the western US) are likely candidates. Addor et al. (2018) discusses this briefly, but much more could be done here.

It would be good to understand this co-variability and modify the discussion accordingly, particularly the conclusions on lines 173-180. Spatial proximity or the attributes defined as climate by the authors are bad predictors in areas with heterogeneous topography precisely because topography and climate are intertwined. That does not mean that climate is a poor predictor of catchment behavior in those same regions.

3) Could more explanation be given as to how the clusters contain basins from very different locations (e.g. cluster 4)? There is some discussion in the appendix, which is good, but this cluster highlights limitations in our current clustering methods or application of those methods. How could other hydrologic signatures be used to provide more discriminatory power? Is predictability in space the best metric to determine which signatures to use in a study like this?

Also, it seems like more discussion on the issues/benefits of using this method (clustering on principle components) using already aggregated data (signatures and catchment averaged attributes) would be useful. This could help the community learn more from these various clustering studies. The authors already provide a summary discussion relating these results to other studies, so I do not feel like this is out of scope.

Minor comments: 1) The sentence starting on line 55 and ending on line 59 is a very

long run-on sentence. It is hard to follow and should be reworked. I suggest checking the manuscript for other instances of run-on sentences.

Figures: 1) Please consider increasing the contrast in the cluster colors in Figures 1 and 3. Specifically clusters 1-3, and 4-6 are hard to visually separate.

Sincerely, Andrew Newman

References:

Addor, N., Nearing, G., Prieto, C., Newman, A. J., Le Vine, N., & Clark, M. P. (2018). A ranking of hydrological signatures based on their predictability in space. Water Resources Research, 54, 8792–8812. https://doi.org/10.1029/ 2018WR022606

Newman, A. J., Clark, M. P., Sampson, K., Wood, A., Hay, L. E., Bock, A., et al. (2015). Development of a large-sample watershed-scale hydro- meteorological dataset for the contiguous USA: Dataset characteristics and assessment of regional variability in hydrologic model performance. Hydrology and Earth System Sciences, 19(1), 209–223. https://doi.org/10.5194/hess-19-209-2015

---

## Author Comment (AC1) · 19 Jun 2019

We would like to thank the reviewers for their constructive comments on the manuscript "Clustering CAMELS using hydrological signatures with high spatial predictability"

(comments of the referees are printed in blue, responses of authors are held in black, added text to the manuscript is in italic)

**Response to Reviewer #1 (Anonymous)**

Jehn et al. classified the CAMELS catchments based on hydrological signatures, and subsequently investigated the link between catchment attributes and the classes. The conclusion of the study is that catchment behavior can mainly be attributed to climate in regions with homogeneous topography, but that this is more difficult in regions with heterogeneous topography. Unfortunately, my perception is that the conclusions of the study are based on a fallacy. The main problem can be found here: "If climate were the main driver, the clusters would be located along a climatic gradient. However, this is only true for the eastern half of the United States (for a climatic map of the United states see (Beck et al.,2018). In this part of the United States, the low relief allows large regions with a uniform climate, that only changes of larger scales." If looking at the map in Beck et al. (2018), but also Peel et al. (HESS, 2007), or Knoben et al. (WRR, 2018), indeed the eastern part of the US shows large regions with uniform climate. But the maps also all show the large scattering in climates in the west: there is more spatial variation in climate in the western part of the US. This therefore seems no justification to state that climate is less relevant in regions with varying topography - there, the climate is just more variable too. This is also confirmed by the results of the study, where precipitation falling as snow is found as one of the main indicators in the west. "This implies that climate is a good indicator for the discharge characteristics as long as the topography is homogenous." seems therefore a too strict and incorrect conclusion, that does not necessarily follows from the results / figures.

After receiving those very constructive comments on our first version of the manuscript, we did a mayor reanalysis of our data and provide additional comparisons and tests. This changed our conclusions and we now have rewritten substantial parts of the paper to accommodate this. Section 3.3 was in the focus of the reviewers concerns and is now changed to:

[revised manuscript text omitted]

*Furthermore, I wonder to what extent 'homogeneous topography' can be found as criterion, when looking at the catchment scale, considering that most catchments in CAMELS are rather small.*

We do not use this phrasing anymore in the revised version of the manuscript.

*Besides my disagreement with the main conclusion, I consider the insights gained from the paper low compared to already available literature, especially considering Addor etal. (2018) and Knoben et al. (WRR, 2018). What did we learn from this study about the relation between attributes and signatures, or catchment clustering, that was unknown before? Especially given that I disagree with the main conclusion. If is it the method applied (PCA combined with clustering), then further elaborate on the methods and better explain everything that is done and how it differs from other studies. This also needs explanation why this method would provide insights that cannot / haven't been obtained with other methods. I would like to encourage the authors to dive deeper into the material and expand the analysis, and have a critical look at their own conclusions.*

To widen the scope of this study and to address the differences in clustering approaches that use hydrological behavior and climate respectively, we added a new section to discuss those topics:

**3.5 Comparing catchment clusters based on hydrological behavior and climate**

*Besides hydrological behavior, climate is often used to sort catchments into similar groups (e.g. Berghuijs et al., 2014; Knoben et al., 2018). Therefore, we are interested if both approaches deliver comparable results. To evaluate this, we contrasted our results to the commonly used Koeppen-Geiger climate classification (Beck et al., 2018) (Figure 7) and recently published approach of Knoben et al. (2018), who sorted climate along three continuous axis of aridity, seasonality and fraction of precipitation falling as snow (Figure 8). The resulting clusters based on climate and hydrology should be the same, if climate is the dominating driver of hydrological behavior in every catchment. Yet, this is not the case for the Koeppen-Geiger classification. In every hydrological cluster are at least two different climates regarding the Koeppen-Geiger classification, ranging up to eight different climatic regions for Cluster 2 and 8 (those even include deserts and very cold regions). Thus, the Koeppen-Geiger classification seems unable to capture the essential drivers of hydrological behavior. A critique also raised in other studies (e.g. Haines et al. (1988); Knoben et al. (2018)).*

[Figure]

**Figure 7: Membership of Koeppen-Geiger clusters (Beck et al. (2018)) in the hydrological clusters.**

*The picture is less clear concerning the climatic index space of Knoben et al. (2018) (Figure 8a). Due to the continuous nature of the approach of Knoben et al. (2018), there are no clear boundaries as in the Koeppen-Geiger classification. Still, there are some emerging patterns.*

*For example, according to the approach of Knoben et al. Cluster 1 is mainly defined by a relatively arid climate, with some seasonal variability and little to no snow. This is in line with our analysis of the most influential catchment attributes for this cluster, as we identified aridity as the main driver. Contrastingly, we could not identify a clear dominating catchment attribute, if we look at Cluster 4 (located in the Northwestern Forested Mountains and Florida) (Figure 5). Catchments with this hydrological behavior can be found in the space of the climatic indices of Knoben et al. with very different aridity, seasonality and fraction of the precipitation falling as snow. There seem to be regions were the forcing signal of the climate is transferred more directly to a streamflow response than in others. However, this does not mean that climate is unimportant in those regions. Either the climate forcing signal is changed more through other attributes of the catchment, or the mean values describing the climate do not properly reflect the variability of the climate in the single catchments. This leads to less clear correlation between the climate and the hydrological behavior. Interestingly, when we look at the single hydrological signatures in the climate index space (Figure 8b, A4) we see a very clear connection between the single hydrological signatures and the climate. This direct connection of the signatures used was also found by Addor et al. (2018). Our results and the comparison show that the complex hydrological behavior, captured in a range of hydrological signatures, does not simply follow the climate only, even though the individual signatures do. This is even more remarkable, as the signatures used are linked to climate directly. For example, the signature "mean half flow date" can be seen as a measure of seasonality. Still, all signatures combined seem to capture a dynamic, which is climatic in origin, but is shaped through the attributes of the catchments (like vegetation and soils (Berghuijs et al., 2014)). Therefore, to find truly similar catchments, using climate characteristics only, is probably not sufficient.*

[Figure]

**Figure 8: a) Comparison of the hydrological clustering of this study with the climate index space of Knoben et al. (2018). Single dots show the catchments and are colored by their hydrological clusters. b) Mean annual discharge for all catchments in the climate index space of Knoben et al. (2018). Single dots show the catchments and are colored according to the value of the mean annual discharge. The log of the mean annual discharge is used to show the relative differences between the catchments. For a depiction of all hydrological signatures used, see Figure A4.**

To reflect the abovementioned changes, we have also rewritten the abstract and the summary and conclusion:

- Abstract

*The behavior of every catchment is unique. Still, we seek for ways to classify them as this helps to improve hydrological theories. In this study, we use hydrological signatures that were recently identified as those with highest spatial predictability to clusters 643 catchments from the CAMELS data set. We analyze the connections between the resulting clusters and the catchment attributes and relate this to the co-variability of the catchment attributes. To explore whether the observed differences result from clustering catchments by either climate or hydrological behavior, we compare the hydrological clusters to climatic ones. We find that aridity is more important for hydrological behavior in the eastern US, while it is the amount of snow in the West. In the comparison of climatic and hydrological clusters, we see that the widely used Koeppen-Geiger climate classification is unsuitable to find hydrologically similar catchments. However, in comparison with a novel, hydrologically based continuous climate classifications, some clusters follow the climate classification very directly, whilst others do not. From those results, we conclude that the signal of the climatic forcing can be found more explicitly in the behavior of some catchments than in others. It remains unclear if this is caused by a higher intra-catchment variability of the climate or a higher influence of other catchment attributes, overlaying the climate signal. Our findings suggest that very different sets of catchment attributes and climate can cause very similar hydrological behavior of catchments - a sort of equifinality of the catchment response.*

- Summary and conclusion

*This study explored the influence of catchment attributes on the discharge characteristics in the CAMELS dataset. We found that over the whole dataset climate (especially aridity) is the most important factor for the discharge characteristics. This changes when we take a closer look at clusters that are derived from specific hydrological signatures. For the clusters in the eastern US, aridity is still the most important catchment attribute. In the western US however, the amount of snow is more important. In addition, in the western catchments the hydrological behavior is less correlated with the remaining catchment attributes. It seems like the clear climatic signal in the east is dampened in the west. This might be caused by a higher influence of other catchment attributes like elevation and vegetation. A similar effect can be found, when we compare how catchment align along hydrological and climatic axes. While some hydrological clusters align along a relatively narrow range of values of the climatic indices, others are found in very contrasting climates. Summarizing, there are differences of how directly the signal of forcing climate can be found again in the hydrological behavior. This explains why catchments often show a surprisingly similar behavior across many different climate and landscape properties (Troch et al., 2013) and why the most hydrologically similar catchment can be hundreds of kilometers away.*

*The aggregated data used in this study might level out the variability of the catchment attributes in the single catchment, but it also indicates that there is a kind of equifinality in the behavior of catchments. Different sets of intertwined climate forcing and catchment attributes could lead to a very similar overall behavior, not unlike to hydrological models that produce the same discharge with different sets of parameters.*

*We acknowledge that the results are dependent on the amount and size of the clusters, the catchment attributes considered and the hydrological signatures used. Still, we think that the CAMELS dataset offers an excellent overview of different kinds of catchments in contrasting climatic and topographic regions. Nevertheless, it seems that even a comprehensive dataset like CAMELS, does not allow an easy way to find a conclusive set of clusters for catchments. For future research, it might be a worthwhile pathway to include measures of spatial variability of the climate in the single catchments. This might help to prove, if a less clear*

*climatic signal is caused by intra-catchment variability of the climate or a larger influence of other catchment attributes.*

---

## Author Comment (AC2) · 19 Jun 2019

We would like to thank the reviewers for their constructive comments on the manuscript "Clustering CAMELS using hydrological signatures with high spatial predictability"

(comments of the referees are printed in blue, responses of authors are held in black, added text to the manuscript is in italic)

**Response to Reviewer #2 (Andrew Newman)**

General comments: This paper examines the CAMELS catchments and clusters them using hydrologic signatures that have been previously found to have high spatial predictability. Overall this study is somewhat unsatisfying. Little new physical insight is gained in understanding how we determine similarity across catchments. The results do agree with past studies, which is a good test of the previous work. However, what does this specific study bring to us? Previous results discussed here, e.g. Addor et al. (2018) (Fig. 4), and Newman et al. (2015) (Fig. 12) found the same results. Aridity is the primary driver of basin behavior given the catchment scale attributes used, followed by other climate indices (e.g. snow). Finally, one of the primary conclusions drawn from the clustering results needs to be reexamined (specific comment #2).

First of all, we would like to thank the Reviewer herein to provide his very constructive comments. We tried to pick up all points, which lead to a revised version of this manuscript, which provides from our point of view now a clearer insight into the gained understanding and the novelty of this research.

Specific comments:
1) Is 10 clusters necessary? Why does 10 make this study similar to others? Did those studies arbitrarily pick 10 also? I wonder if a similar cluster selection method is better, rather than the same number of clusters. A more detailed justification in the methods section is necessary.

To further elaborate on our choice of 10 clusters, we added a more detailed explanation of our decision in section 2.3:
*From those studies, Kuentz et al. (2018) provides the largest set with over 35,000 catchments. They also clustered their catchments in a PCA space of a range of hydrological signatures. To select the number of clusters, they used the elbow method (and two other methods to validate their results) and found that ten or eleven clusters (depending on the method) were most appropriate for their data. Due to the similarity in the clustered data and the larger database of Kuentz et al. (2018), we also used ten clusters.*

2) Many of the attributes have high co-variability. For example, elevation and temperature/fraction of snowfall, elevation and mean slope, forest fraction and elevation (in the western US) are likely candidates. Addor et al. (2018) discusses this briefly, but much more could be done here. It would be good to understand this co-variability and modify the discussion accordingly, particularly the conclusions on lines 173-180. Spatial proximity or the attributes defined as climate by the authors are bad predictors in areas with heterogeneous topography precisely because topography and climate are intertwined. That does not mean that climate is a poor predictor of catchment behavior in those same regions.

In light of the reanalysis of our data, we have mostly rewritten section 3.3 and added a discussion of the co-variability of the catchment attributes. This also changes our discussion of the connection between the topography and the climate.

[revised manuscript text omitted]

3) Could more explanation be given as to how the clusters contain basins from very different locations (e.g. cluster 4)? There is some discussion in the appendix, which is good, but this cluster highlights limitations in our current clustering methods or application of those methods. How could other hydrologic signatures be used to provide more discriminatory power? Is predictability in space the best metric to determine which signatures to use in a study like this?

This is now discussed in section 3.3:

*[…]*

*This indicates that clustering by using spatial proximity might only work in regions like in the eastern US, where the behavior of rivers changes gradually. The finding that the most similar catchment (based on their hydrological signatures) can be far away, also explains the behavior of clusters that contain catchment quite far away from each other (e.g. Cluster 4). The catchments might be far away from each other, but the interplay of different catchment attributes and driving factors can lead to similar discharge behavior*

*[…]*

Also, it seems like more discussion on the issues/benefits of using this method (clustering on principle components) using already aggregated data (signatures and catchment averaged attributes) would be useful. This could help the community learn more from these various clustering studies. The authors already provide a summary discussion relating these results to other studies, so I do not feel like this is out of scope.

We added a short discussion of this to section 3.4:
*In addition, this study shows that using clusters derived from principal components of hydrological signatures create meaningful groups of catchments with similar attributes (Figure A2, A3). Those clusters also show distinct spatial patterns (Figure 3). Similar results were also found in other studies that used the same method (Kuentz et al., 2017; McManamay et al., 2014), but based them on partly different hydrological signatures. Therefore, the principal components of hydrological signatures can be used as a measure of similarity between catchments. They represent the "essence" of all hydrological signatures used. Our results also show that it is difficult to link those catchment clusters to simple averaged measures of catchment attributes. While some clusters have very clear connections to the attributes, others have no catchment attribute that could easily explain the behavior of the catchments. This hints, that some catchments are easier to explain (in a hydrological sense) than others. Those difficulties might be an artifact of the averaged catchment attributes or be caused by complex catchment reaction, forced by intertwined climate and catchment attributes. Which in turn, might indicate an equifinality of catchment response.*

Minor comments:

The sentence starting on line 55 and ending on line 59 is a very long run-on sentence. It is hard to follow and should be reworked. I suggest checking the manuscript for other instances of run-on sentences.

Changed as proposed.

Figures: 1) Please consider increasing the contrast in the cluster colors in Figures 1and 3. Specifically clusters 1-3, and 4-6 are hard to visually separate.

Changed as proposed. We changed all figures with clusters to a more easily distinguishable color scheme. We also changed Figure A2 and A3 from swarm plots to violin plots, to make them easier to interpret.

---

## Author Response (AR2)

We would like to thank the reviewers for their constructive comments on the manuscript "Using hydrological and climatic catchment clusters to explore drivers of catchment behavior."

(comments of the referees are printed in blue, responses of authors are held in black, added text to the manuscript is in italic)

**Response to Reviewer #1 (Anonymous)**

Jehn et al made substantial improvements to the manuscript based on the first review round. Analyses were added, and the results are now better embedded in the literature and compared to existing classification strategies. There is, however, still one question that remains after reading the manuscript, which I will discuss below.
Major:
The added analyses provide valuable glimpses on what exactly explains the differences among the different catchment clusters, but it does not become exactly clear what characterizes each cluster. This is of course the result of the approach taken; rather than classifying starting off from process-understanding (or expert opinion, if you will), the authors chose to start off with just the data, and from there derive clusters using PCA, subsequently trying to characterize the found clusters. Although I don't think this approach is by definition is invalid (i.e., in principle nothing is wrong with this approach), for me as a modeler frequently working with large sample data, I am not sure if I would use the classification presented because I wouldn't know how to interpret the subsequent results.
For large sample studies, catchments often need to be clustered because different types of catchment behavior are not equally represented in the large-sample data set. Therefore, catchment classifications are highly valuable. But how you will classify the catchments depends on the research question. If aridity is expected to have a big impact on what is studied, it would perhaps be better to cluster the catchments on aridity, if snow is of substantial importance for the process studied, the catchments can be clustered on their snow fraction.

As you stated, it is impossible to find a set of clusters that can be used for all applications and indeed, if you are only interested in aridity or snow, the clusters discussed in this study here will probably be of little use. However, our study does not claim to provide clusters that can be used in every case. Rather, we created clusters based on hydrological signatures with a clear hydrological meaning as discussed by Addor et al. 2018. Doing so, we capture the overall hydrological behavior of the catchments. An example use case for such clusters would be the research into parameter transferability between catchments, as those rely on hydrological similarity between catchments. As of now, Jan Seibert is using the clusters discussed in this study (and other metrics) to transfer parameters between catchments.
The main goal of our study is of a more exploratory nature. It is often assumed that catchments are mainly (or even solely) driven by their climatic forcing. However, if this were the case, using hydrologically derived clusters, should always result in a uniform climate in those clusters. We challenged this assumption by clustering according to hydrological similarity and then exploring a) which catchment attributes seem to be the most important one in each cluster and b) how those clusters can be found again in a climatic clustering. As shown by our study, climate does not seem to be the single driver for all catchments and we probably should take a deeper look on how the climatic forcing is shaped by the catchment attributes.
To make this intent of our study more clear, we added the following to the introduction (before the last section of the introduction):

*In addition, if the climate is the dominant driver of catchment behavior, clustering catchments based on their hydrological behavior should result in clusters with a uniform climate.*

The point is that for the classification presented in the manuscript, it is unclear where they are clustered on, and therefore the application for large sample studies remains unclear.

We have reorganized and partly rewritten section 2.2 Data analysis, to make it clearer what the clusters are based on. The whole section now reads as follows:

*2.2 Data analysis*

*The workflow of the data analysis considers a data reduction approach with a principal component analysis and a subsequent clustering of the principal components, similar to Kuentz et al. (2017) and McManamay et al. (2014). For the principal component analysis and the clustering, we used the Python package sklearn (0.19.1). The code is available at GitHub (Jehn, 2018). Validity was checked by a random selection of 50 and 75 % of all catchments. We found that the overall picture stayed the same (not shown). In all further analysis, we used all catchments to get a sample as large as possible to be able to make statements that are more general.*

*Calculation of the principal component analysis*

*The principal components were calculated from the six hydrological signatures described above (Table 1). We used a principal component analysis on the hydrological signatures to remove correlations between the single hydrological signatures. We only used principal components that together account for at least 80% of the total variance of the hydrological signatures, which resulted in two principal components. Those two principal components contain the uncorrelated information of all hydrological signatures used and thus can be seen as describers of the overall hydrological behavior. Therefore, catchments with similar principal components have similar hydrological behavior.*

*Evaluating the connection between the principal components and the catchment attributes*

1) *First, we calculated quadratic regressions between the two principal components and the catchment attributes (with the principal component as the dependent variable). This resulted in one coefficient of determination ($R^2$) for each pair of principal component and catchment attribute (e.g. PC 1 and aridity).*
2) *We then weighted the $R^2$ by the explained variance of the principal components. This addresses the differences in the explained variance of the principal components (e.g., PC 1 explained 75% of the variance, PC 2 explained 19% of the variance).*
3) *The weighted coefficients of determination of the principal components were subsequently added to obtain one coefficient of determination for every catchment attribute.*

*Quadratic regression was selected as interactions in natural hydrological systems are known to have unclear patterns and can therefore often not be fitted with a simple straight line (Addor et al., 2017; Costanza et al., 1993). This was done first for the whole dataset and then for all clusters separately. This procedure captures the pattern on the catchment attributes in the PCA space of the hydrological signatures (for examples of this pattern see Figure A1).*

*Clustering the principal components*

*The principal components of the hydrological signatures were clustered following agglomerative hierarchical clustering with ward linkage (Ward, 1963), similar to previous studies (Kuentz et al., 2017; Li et al., 2018; Yeung and Ruzzo, 2001). From those studies, Kuentz et al. (2018) provides the largest set with over 35,000 catchments. They also clustered their catchments in a PCA space of a range of hydrological signatures. To select the number of clusters, they used the elbow method (and two other methods to validate their*

*results) and found that ten or eleven clusters (depending on the method) were most appropriate for their data. Due to the similarity in the clustered data and the larger database of Kuentz et al. (2018), we also used ten clusters. Those ten clusters represent groups of catchments with distinctly different hydrological behavior.*

It would be really helpful if every cluster has a clear signature or characteristic that distinguished it from other clusters, and as such would ease the interpretation of large-sample study results. Sort of related to this point is the choice of the six hydrological signatures. It is clear why these were selected (Addor et al), but question remains how different the clusters would be if different / fewer signatures were used. Related to the previous point; if one is interested in high flows, this researcher might prefer a cluster solely based on the 95% flow quantile. Another point, remarked by the authors, is that a low-flow signature is not accounted for, for reasons explained in Addor et al., but that makes again the results more difficult to interpret – do the clusters represent the flow with focus on higher discharge ranges?

The clusters do have typical signatures and characteristics. They are described in Table 2 and to more depth in the appendix. It is true though, that those clusters are somewhat fuzzy in their membership. However, this is exactly one of the sticking points of our study: If we use the hydrological signatures described as most hydrologically meaningful and we still are not able to create crisp clusters, than this means in turn that solely relying on climate or single catchment attributes to cluster catchments is not a reliable way to find similar catchments. We also stress this and the choice of the hydrological signatures in the last section of the conclusion.

In other words, for which type of studies should this classification be used?

As mentioned above, the classification is for exploratory purposes and we test the several aims outlined at the end of the introduction. Still, possible use cases for this clustering are testing hydrological models in contrasting environments with a focus on either parameter transferability or testing different model structures and their transferability. As each cluster represents catchments that have similar behavior over a range of hydrological signatures and thus hydrological behavior, parameters and model structures should be transferable. As discussed in the conclusions, the results of this study can also be used to point to further research. We think it is quite worthwhile to explore if a less clear climatic signal is caused by intra-catchment variability of the climate or a larger influence of other catchment attributes.

Minor:
p.11 l.220: 'We link this to a less strong climatic signal in those regions' – it is unclear to me what you mean by this. Please clarify / reformulate.

We reformulated the sentence:

*We link this to the signal of the climatic forcing being more influenced by other catchment attributes, which results in a less clear connection between hydrological behavior and climate.*

p.17 l.301: 'For example, the signature 'mean half flow date' can be seen as a measure of seasonality' – I disagree with this statement. This for instance of a catchment with the same flow throughout the whole year, the half flow date will be in the middle of the year. Now think of a catchment with a very clear seasonality and the discharge peak halfway the year; also here the half flow date will be in the middle of the year. Half flow date does not say anything

about the variation over the seasons. Therefore, I believe that the last part of section 3.5 requires reformulation.

We removed this statement and the sentence that refers to it.

Although the manuscript generally reads well and has a nice flow to read, there are some minor typo's/language edits that can be resolved. Small example of typo and language edit is p.19 l.317 last word ('catchment') should be plural, and the 'Still' (p.19 l.328) does not precede a 'Nevertheless' (p.19 l.329) really well.

Changed as proposed.

Summarizing, my main issue is that it remains unclear how the classification should be used and can help in interpreting large-sample studies. If the authors are able to clarify this, it would be nice if the cluster numbers along with the CAMELS catchment ID would be shared publicly for others to use.

The cluster numbers together with the catchment IDs and the climatic indices are already available in the supplement of this paper. To make this clearer we now refer to it in the data availability section.

**Response to Reviewer #2 (Andrew Newman)**

I appreciate the extensive revisions to this article. I believe the authors have taken the reviewer comments to heart and the revised article is much improved. The comparisons to the Köppen-Geiger climate classes and the Knoben et al. (2018) indices are interesting. There are a few remaining errors and clarifications that need to be addressed.

Specific comments:

Line 176: "including obviously different climates". I think the opposite is also true: part of the reason basins that are very far away end up in the same cluster is that they may actually have very similar climates. Of course the authors' statement is also true, sometimes basins with different climates have similar hydrologic signatures.

We replaced "obviously" with "sometimes very" to emphasis this.

*Even though the catchments might be far away from each other, the interplay of different catchment attributes and driving factors, including sometimes very different climates, can lead to similar (equifinal) discharge behavior.*

Line 291-293. It is unclear to me what exactly these lines are referring to. Is it discussing clusters that have weak or strong climate signal in the attributes?

Those lines discuss that clusters where we could identify the most important catchment attributes, where also those that tend to have a clearer pattern in the climate index space. In contrast, clusters without a clearly identifiable most important catchment attribute, have catchments in all regions of the climate index space. We rephrased the section to:

*This is in line with our analysis of the most influential catchment attributes for this cluster, as we identified aridity as the main driver. Contrastingly, clusters where we could not identify a clear dominating catchment attribute, e.g. Cluster 4 (located in the Northwestern Forested*

*Mountains and Florida) (Figure 5), also have no clear clustering in the climate index space. The catchments of those clusters can be found in the space of the climatic indices of Knoben et al. (2018) with very different aridity, seasonality and fraction of the precipitation falling as snow.*

Figure 8. In the panels, is the aridity label flipped? It seems that basins with high daily flow or wet clusters (e.g. cluster 6) end up with negative values, or more arid? That seems incorrect.

We double-checked the figure and it is correct. The position of e.g. cluster 6 on the aridity scale is probably caused by the seasonality and the way the aridity is calculated. Knoben et al. (2018) calculate a moisture index for every month separately and then take the mean of all months as a measure of aridity. The catchments in question probably have mostly arid month and then some with quite a lot of precipitation. Cluster 6 for example has very low mean summer discharge, but quite high mean winter discharge. In addition, the mean discharge is sensitive to extreme values in the high flows, which seems to be more common in the more arid catchments.

[revised manuscript text omitted]

---

## Author Response (AR3)

We would like to thank the reviewer for the very helpful and constructive comment on the manuscript "Using hydrological and climatic catchment clusters to explore drivers of catchment behavior".

(comments of the reviewer are printed in blue, responses of the authors are held in black, added text to the manuscript is in italic)

**Response to Reviewer #1 (Anonymous)**

Summary

The authors attempt to address the question: "where is hydrologic behavior similar across the contiguous United States?" They use Principal Component Analysis and quadratic regression to cluster catchments in the CAMELS data set located in the Contiguous United States. The variables of interest are 6 hydrologic signatures that earlier research has shown to have high spatial predictability for this dataset. The authors use 15 catchment attributes that were shown to strongly correlate to these 6 signatures to explain the generated clusters in terms of catchment similarity. They discuss which attributes are most influential in determining each cluster and take some steps towards interpreting this from a hydrological processes point of view.

I have read this paper with great interest. I find the separate correlation plots between the east and west CONUS (Figure 6) very interesting and am curious about the differences in eco-regions and hydrologic behavior between the east and west that this might imply.

We are very happy with the interest of the reviewer in our manuscript and like the idea of including the ecoregions in Figure 4. As it turned out, there is a strong correlation and we decided to change the background of the cluster map to the level 1 ecoregions of the Environmental Protection Agency.

[Figure]

**Figure 4: Locations of the clustered CAMELS catchments and level I ecoregions (Omernik and Griffith, 2014) in the continental US. Dotted line marks the 100th meridian.**

To put a larger emphasis on the correlation plots between the east and west CONUS we changed them from Figure 6 to be Figure 1. Additionally we added a chapter at the beginning of the results and discussion, where we describe the correlations between the catchment attributes and the ecoregions. We also include the precipitation seasonality as an additional catchment attribute in this figure. This section reads as follows:

*3.1 Catchment attribute correlations in the CAMELS data set*

*Usually the 100th meridian is seen as the dividing climatic line in the US, splitting the country in a semi-arid west and a humid east. We assume that this difference in climate also has implications for the hydrology and the overall catchment attributes in those regions. To quantify this we split the CAMELS data set into a western and an eastern part, based on the 100th meridian (Figure 1 and 4). This shows that many of the catchment attribute correlations do not differ much between the east and the west. In most cases (>80%), Pearson correlation coefficients vary by less than 0.4 (Figure 1c). Still, there are some catchment attributes with larger differences of up to 0.7 between both regions. Most striking are the mean elevation and the fraction of the precipitation falling as snow as well as the vegetation attributes LAI maximum and Green vegetation fraction maximum. Even though these attributes are directly related to each other through temperature gradients, they differ substantially in both parts of the country. In the mountainous western US, elevation is highly correlated with the fraction of precipitation falling as snow (r=0.8), while it is not in the eastern US (r=0.4). This, and the different correlations between vegetation and elevation are probably caused by the fact that the temperature gradients differ in both regions. In the western US it is much more mountainous and thus temperatures typically change with elevation. In the more level eastern US, on the other hand, the change in temperature is mainly linked to the latitude. Striking are also the changes of correlation with regard to the fraction of precipitation falling as snow. Here we find altered directions of the correlation, i.e., positive correlations with LAI maximum and frequency of high precipitation events in the east turn to negative ones in the west. It also becomes obvious that all three measures of vegetation seem to track similar characteristics in the catchments, as they highly correlate with each other (especially in the eastern US with r=0.9). In addition, all vegetation attributes depict a large negative correlation with aridity. Hence, the vegetation attributes considered are likely good proxies for aridity. Overall, we see that the relations between the catchment attributes are quite similar for the eastern and western US, with the exception of the mean elevation, snow and the LAI maximum.*

[Figure]

**Figure 1: Pearson correlation coefficients given for all catchment attributes in western (a) and eastern (b) US. Absolute differences of the correlation coefficients between the eastern and western US is given in c). Eastern and western is defined by the 100[th] meridian. Due to rounding effects, correlations with the same Pearson correlation coefficient might show slightly varying color codes.**

My main concern is that apart from Figure 6, the manuscript mostly seems to confirm earlier work by e.g. Addor et al. (2018), Berghuijs et al. (2014), Knoben et al. (2018) and Kuentz et al. (2017). Confirming findings is not a bad thing, but I think the authors are missing out on an opportunity to go beyond these studies. The authors spend some time in the main manuscript (L219-229; L256-258; Table 2 to some extent) speculating about hydrologic behavior in each of their clusters. More of these thoughts are hidden in the appendices (L460-511). I believe the manuscript would become much stronger if the authors would make this the main topic of the manuscript and spend more time on trying to understand the hydrologic behavior each cluster represents in terms of dominant processes, as this would be a novel contribution to the field. This could be structured similar to

Berghuijs et al. (2014) but the CAMELS dataset gives the authors the catchment information needed to go beyond that work. Addor et al. (2018) could also help to outline potential changes to the manuscript.

The reviewer is right here. An in depth analysis of the processes within the clusters was missing so far. In the revised version, we added a new chapter where we discuss the clusters with their processes in depth:

[revised manuscript text omitted]

I also have a few methodological concerns about the way the signatures and attributes have been selected and how the number of clusters has been determined, and how these choices might limit the authors' ability to go beyond these earlier studies (see below).

Major comments

1. The authors use the six most predictable signatures from Addor et al. (2018) for analysis here. They use 15 catchment attributes that Addor et al. (2018) indicates as having the highest ability to explain these signatures, with climatic attributes having the strongest connection to signatures. Earlier work (Berghuijs et al., 2014; Kuentz et al., 2017; Knoben et al., 2018) has also shown the strong influence climatic conditions have on hydrologic behavior and advocate for further studies

that investigate the impact of less clear relations between catchment attributes, such as resulting from geology or vegetation, and hydrologic behavior. These relations can be seen in well-monitored experimental catchments and must logically exist in all other catchments, but in large-sample studies this has so far not been conclusively shown (Addor et al., 2018, provides various compelling reasons for why that might be the case). Progress towards understanding these relations would be an important contribution to the literature and interpreting cluster analysis from a dominant hydrological process perspective could be a first step. The authors take some steps in this direction, but unfortunately they are limited by their study setup to mostly confirm what has already been shown before, without having the necessary information available to go beyond these earlier studies. Restructuring of some of the study setup and analysis might be needed (see points 2 and 3 below).

We agree and have included a more complete description and interpretation of our clusters in section 3.4, see previous comment. In addition, we have rewritten section 3.5:

*3.5 Importance of the catchment attributes in the clusters*

[revised manuscript text omitted]

2. Hydrologic interpretation is limited by the choice of signatures. I appreciate that the authors were looking for signatures that are easily predictable in space, but this limits the generality of the conclusions that can be drawn. The chosen six signatures do not describe the full hydrologic regime, focusing mostly on flow magnitude (mean annual, summer and winter flow, runoff ratio, Q95) and somewhat on seasonality (half-flow date), with no signatures dedicated to low flows, intermittency of flows, or response time of the catchment. Therefore statements such as the following are too general for the supporting analysis and should be rephrased to account for the specific conditions these 6 signatures describe (please note that this list could be incomplete):
- L95. "These two principal … overall hydrological behavior."
- L188. " … lead to similar (equifinal) discharge behavior"
- L464. "So over one third of the catchments in CAMELS show a relatively similar behavior."
- L470. "… catchments with very different attributes can produce very similar discharge characteristics, …"
- L480. "an example of different catchment attributes being able to create similar discharge characteristics concerning their signatures, while having different catchment attributes"

We changed those statements to make clearer that they refer to the signatures used in this study.

Related to this, both Addor et al. (2018) and Knoben et al. (2018) show that these particular signatures correlate strongly with climatic conditions in the catchment. I doubt whether there is much to be learned about the influence of non-climatic attributes on hydrologic behavior by looking only at signatures with such strong connections to the prevailing climate. Using a wider range of signatures could allow more in-depth analysis of the relation between attributes and signatures. E.g. McMillan et al. (2017) could be of use in choosing different signatures:

McMillan, H., Westerberg, I., & Branger, F. (2017). Five guidelines for selecting hydrological signatures. Hydrological Processes, 31(26), 4757–4761. https://doi.org/10.1002/hyp.11300

We agree that a different set of hydrological signatures might lead to different sets of clusters. However, we did focus on the signatures, as we wanted to know if the signatures identified by Addor et al. (2018) can be used to create meaningful hydrological clusters, which is part of the main research question in this manuscript. Changing the signatures now would lead to an entirely different paper. Consequently, we removed statements in the manuscript, which might have implied a too general picture of the captured hydrological behavior.

Despite not re-clustering with additional signatures, we picked up the idea of investigating the clusters performance with regard to low flows. The new figure 6 provides information how the different discharge patterns are captured by our clusters. Interestingly, it turns out that the catchment's behavior during low flow conditions is very similar in the individual clusters, although we have not included signatures that are concerned with low flows. We conclude that the hydrological signatures we used contain already sufficient information to present in the discharge patterns.

3. Hydrologic interpretation is also limited by the choice of attributes, because the selected attributes are strongly correlated with one another. I had already written a few comments on this before reaching Figure 6, which shows that the authors are aware of these correlations. This knowledge should play a much larger role in the earlier parts of the paper, where the study setup is decided (i.e. which attributes to use) and where the importance of attributes for clustering is discussed (for example, the 5 most important attributes in cluster 3 are essentially 2 factors spread out over 5 attributes: snow & elevation are the first (r = 0.8), and various aspect of vegetation are the second group (r=0.7, r=1 and r =0.8). A different selection of attributes might be needed. I also believe that enforcing 3 attributes per attribute category is unnecessarily limiting and ignores some of the current understanding of drivers of hydrologic behavior, such as not using a climate seasonality metric (further details below).

In line with comments and replies above, we now focus on the former figure 6 (which is now figure 1) in this manuscript. A new in-depth analysis on the correlations in the data set is given (section 3.1). Further, we include the precipitation seasonality as a climate seasonality metric that is provided in the CAMELS dataset.

Further missing catchment attributes, which are present in the CAMELS dataset, such as the fraction of carbonate rocks or the water fraction in the soil, have not been mentioned in any of the literature we have reviewed as being very important for hydrological behavior.

Minor comments

L45. Addor et al. (2018) identified these signatures as having low spatial predictability in the US. Is it correct to assume that these conclusions also apply to the study domain of Kuentz et al. (2017), i.e. Europe?

To be honest, we do not know. Obviously, Europe and the continental US are different in many aspects, but we would be surprised if they are so different that hydrological findings cannot be transferred between those regions, especially when they are derived from large data sets.

L68. Is there a reason to assume that the most diverse total information is retained by using 3 attributes each from climate, topography, vegetation, soil and geology? To what extent are all

CAMELS attributes correlated and to what extent is the subset of 15 attributes correlated with one another?

This point is captured in our answer to the main point 3 of the reviewer.

L69. [Adding to the previous comment] Among others, Berghuijs et al. (2014), Addor et al. (2018) and Knoben et al. (2018) have found that climatic seasonality is an important control on hydrologic behavior. The authors have included 'frequency of high precipitation events' over a climate seasonality metric. I agree that there can be good reasons to include the frequency of high P events metric but because the authors limit themselves to 3 attributes per attribute category, they cannot include a seasonality attribute even though current theory indicates that seasonality can be an important control on hydrologic behavior. Given this, I think the choice of 15 attributes and how they are distributed between the different categories needs to be better justified and possibly changed.

As suggested by the reviewer, we included precipitation seasonality in our analysis and the revised version of the manuscript. See further replies to this issue in the main points raised.

L95-97. I don't think the two PCA's of the six signatures can be seen as "describers of the overall hydrologic behavior". This sentence and the next one need to be more nuanced, because the authors state in section 2.1 that no low flow signatures are part of their selection. Other possibly relevant aspects of the flow regime, such as baseflow or flashiness, are also not covered in this selection of signatures.

Changed as proposed.

L119. Kuentz et al. (2017) use 10 clusters to group >35000 catchments using 16 different signatures. I expect that choosing to use 10 clusters in this study with >600 catchments and 6 signatures might provide unnecessary granularity. Can the authors somehow quantify the difference between each pair of clusters to show that 10 is an appropriate number? If such quantification is not possible, did the authors investigate the impact of using fewer or more clusters?

[additional note] Seeing that cluster 3 only contains 7 catchments and that cluster 5 only has 9, but that cluster 1 has 230 catchments in it, I think that some more discussion of the number of clusters is warranted. Cluster 5, 6 and 7 also look very similar, possibly indicating that too many clusters have been used. Some questions that come to mind:

- What is the explanatory power of a cluster with only a handful of catchments in it?

We also tried to use the elbow method (that was used by Kuentz et al (2017)) to find the right amount of clusters. However, this did not produce a clear cut answer on how many clusters should be used in our study. Therefore, we assumed that the larger database of Kuentz et al (2017) gives a more reliable estimate of the right amount of clusters. In addition, Berghuisj et al (2014) also found that 10 clusters are a good number to capture the differences in hydrology in the continental US. However, this number of clusters still remains discussable, which is why we provide in the revised version of the manuscript the choice of our number of clusters (section 3.4).

- Is the distribution between clusters so skewed because the catchment sample is not uniformly distributed across the selected attributes?

We think the skewed distribution can be explained quite well with the distance in PCA space: The small clusters are just very different to the bigger clusters (Figure 3). We see no reason, why hydrological behavior in rivers should be uniformly distributed. Many other kinds of natural phenomenon are shaped in bell curves. For example the size of humans is based on a large set of factors and follows a bell curve. The extreme ends of the curve can only be reached if many factors align in the same direction. This is probably true for rivers as well. "Normal" behavior can probably be reached by all kinds of combinations of catchment attributes, while more extreme behavior needs to have several attributes that force it into that direction.

- Would more and/or different attributes provide more balanced clustering results?

The clustering is not based on the catchment attributes, only on the hydrological signatures.

- If the catchment sample is not uniformly distributed across attribute space, does this influence the PCA results?

The attributes do not affect the clustering and with this also not the PCA results.

L126. That aridity and forest fraction score highest could possibly relate to the high correlation between these two attributes. Investigating the correlations between the 15 catchment attributes could show how much independent information is contained in each. The same could be said about fractional snowfall and elevation. – Note: upon further reading I see that these correlations are in Figure 6. This information should be part of the text here.

We now discuss the correlations earlier (see answer to main point 1).

L137. I don't think calling these six signatures "more hydrologically meaningful" is supported by the findings of Addor et al. (2018). "more gradually varying in space" perhaps.

Changed as proposed.

L144. "This can probably be extrapolated to most catchments in the continental US without human influence, as the CAMELS dataset contains large samples of undisturbed catchments". This sentence is speculation and should be removed. If the authors want to keep this statement it could for example be supported by calculating the climate attributes used by Knoben et al. (2018) and comparing these to the range of values for these attributes found across the CONUS. This would show how climatically representative the CAMELS catchments are for the wider CONUS.

We removed this sentence.

L185. See also Berghuijs et al. (2014) who find hydrologic similarity across comparable distance in the CONUS; or Kuentz et al. (2017) who find hydrologic similarity across comparable distances in Europe;

or Knoben et al. (2018) who find catchments with similar hydrologic regime on different continents, using only climate indicators to describe similarity.

We reference those studies now and mention their similar findings.

L189-195. I suspect that if correlations between attributes are taken into account, many of the attributes that are of high importance in each cluster turn out to be quite directly related one another. For example, (cluster 1) high aridity and low forest fraction & green vegetation fraction maximum will be inversely correlated; (cluster 3) precipitation and snow and elevation will be correlated, as will forest fraction and LAI maximum and green vegetation fraction maximum. Therefore I expect that this part of the analysis will be more instructive if these correlations are accounted for, either in selection of the attributes or by lumping correlated attributes into groups in some fashion. Changes to Figure 5 might be needed.

We discuss the correlations now at several places throughout the manuscript.

L214. "While aridity … single clusters (Figure 5)." Implying that aridity is not important in most of the clusters seems a bit of a stretch. Aridity is the most important attribute in 4 out of 10 clusters, and the second-most important in another 2. It appears in the top 5 of important attributes in 8 out of 10 clusters (and in the remaining 2 clusters the correlated forest fraction appears), more often than any other attribute.

We have rewritten this whole section (3.5), see earlier answers.

L248. "Therefore, our selection of hydrological signatures seems to allow a better identification of hydrological similarities." Unfortunately I think this argument can be reversed as well, in the sense that this selection of signatures might not capture enough of the details of the individual regimes to give the clustering approach any trouble. Because these 6 signatures are strongly related to climate (e.g. Addor et al., 2018; Knoben et al. 2018), and the relevant climate indices are (mostly) included in the clustering approach, it is not surprising that these signatures cluster easily. The fact that the authors don't use a climate seasonality attribute, which has been shown to be an important driver of hydrologic differences, could potentially explain why their Cluster 2 does not seem to have any distinct character. Instead of making this statement and moving on, a strong contribution would be if the authors can determine how to make hydrologic sense of all the catchments that don't seem to follow any obvious pattern. Would different attributes solve this?

We included the climate seasonality metric given in the CAMELS data set and discuss it in sections 3.4 and 3.5. This improved the distinctions for some of the cluster. However, for the mentioned cluster 2 the climate seasonality did not provide much additional information.

L301. I'd argue that Cluster 4 seems to be firmly placed in the non-arid & snow-dominated region of the climate space. There are more catchments in this climate region that belong to different clusters but this is (1) inherent to imposing binary boundaries (catchments are either cluster X or Y, even if they are 49% similar to X and 51% similar to Y) and (2) because the climate plots in Figure 8 only look at a limited selection of possibly influential attributes (climatic or otherwise).

The reviewer is right. Accordingly, we removed the sentence.

L310-315. This connection between signatures and climate can also be seen in Knoben et al. (2018) and Kuentz et al. (2017). Addor et al. (2018), Knoben at al. (2018) and Kuentz et al. (2017) (among others) acknowledge that using climate alone is not sufficient to produce a catchment classification system. This should probably be mentioned as part of this section (or in the introduction of the paper, because it provides a compelling reason for investigating catchment attributes).

We reference this now at the end of the mentioned chapter.

Figure 8. Is the aridity axis upside down in these plots? More arid catchments seem to have higher flows.

It seems that we accidently switched the sign of the aridity values during the extraction from the climate maps of Knoben et al. (2018). We fixed this and the figure should be correct now.

Figure A2. I like the way violin plots look, but kernel density smoothing does not respect physical boundaries very well and distorts the data being plotted. See for example cluster 3 and the mean winter discharge signature, which is, according to the violin, a negative flux for some of the catchments in this cluster. Histograms or box-and-whisker plots would more accurately reflect the data.

As recommended by the reviewer, we now display this information as boxplots.

Figure A3. See comment above.

Changed as proposed.

Figure A4. Is the aridity axis upside down in these plots? More arid catchments seem to have higher flows.

Fixed.

Typographical

30. "those" > "this"?

119. Kuentz et al. (2018) > Kuentz et al. (2017)

P11. Caption of Figure 5. "For the catchment clusters." should not be a stand-alone sentence.

215. "single" > "individual"?

251. I understand what this sentence is meant to say but it doesn't quite work. Is "This human influence might mask otherwise apparent patterns." better?

261. "have" > "has"

265. "cluster" > "clusters"

All changed as proposed.

In addition to the changes mentioned above, we also rewrote parts of the abstract and the summary to accommodate for the changes made on the manuscript.

[revised manuscript text omitted]

---

## Author Response (AR4)

We would like to thank the Anonymous Referee #3 for the very helpful and constructive comment on the manuscript "Using hydrological and climatic catchment clusters to explore drivers of catchment behavior". This has been one of the most productive review processes we have experienced so far!

(comments of the reviewer are printed in blue, responses of the authors are held in black, added text to the manuscript is in italic)

**Response to Reviewer #1 (Anonymous)**

The authors have responded in great detail to the raised concerns and I think the manuscript has improved substantially as a result. The authors now go beyond clustering and try to explain these clusters from a hydrologic point of view and relate them to ecologic regions across the US.

I have outlined several comments that could help clarify the paper further. None of these comments should lead to substantially more work. As a general note, I think all the information is currently there but the manuscript might benefit from a final read-through that focusses on language and communication of the results. Some sentences do not flow very well and some bits of information are replicated several times across the document. I think the manuscript will be better received (without implying that it won't be in its current shape) if the text were a bit more streamlined.

**Comments**

55. "In addition, if ... a uniform climate". This seems an oversimplification. If climate is dominant, cluster could be expected to orient mostly along climate gradients. Expecting uniform climates in the clusters is a bit unrealistic.

Changed to "...with a similar climate."

L96. "Validity was checked ... of all catchments." Can the authors provide a few more details of what validity means in this context and how they determined whether the clustering was valid?

L96. "We found that ... same (now shown)." It might be good to clarify that this tests the effect of random processes in the PCA and clustering analysis (I think).

We also clustered a random selection of 50 and 75 % of the CAMELS dataset. This was done to make sure that our clustering captures a real trend in the underlying data. To make this more clear we rephrased this.

Validity was checked by also clustering a random selection of 50 and 75 % of all catchments. This showed that the clustering stayed the same, independently of the amount of catchments used (not shown).

L140. "Pearson correlation coefficients". I see that the type of correlation wasn't mentioned in the first manuscript I reviewed. Pearson looks for linear correlations. Is there a reason to assume that correlations between attributes are linear? I would be interested to see the same plot (Figure 1) with Spearman rank correlation coefficients. I expect the changes will not be large, but it would be good to have this confirmation.

Spearman is indeed the more sensible choice and we updated the figure in the manuscript accordingly. As expected, there are only minor changes.

**L149. "Here we find ... in the west." Is there a physical mechanism that explains these inverse trends?**

We added an explanation for the LAI.

The change in the LAI maximum might be linked to the higher elevations in the west. In higher elevations less vegetation is growing, but more snow falls.

**Figure 1. It might be clearer to mask the parts of each sub-figure that duplicate information.**

We tried this in earlier versions of our manuscript (before uploading it to HESS), but we had the impression that this makes it harder to understand the figure. Therefore, we would like to keep the figure this way, although the information is partly duplicate.

L166. "Attributes related to ... the highest scores." It might be good to emphasize here that aridity and forest fraction show high correlations in both the east and west.

We added a sentence to clarify that.

However, it should be noted that all vegetation catchment attributes show a strong correlation with the aridity (Figure 1) and thus capture similar trends, in both, the east and the west.

L183. "..., which seems to ... in our analysis." I don't think this is quite true. The attributes "seasonality of precipitation", "precip as snow", "green vegetation fraction maximum" and "LAI maximum" are all indicative of strong seasonal cycles, and these attributes are all in the top 7 important attributes the authors are currently discussing.

This was a misunderstanding due to a not very clear description in the previous version of the manuscript. We rephrased this sentence to make our intended meaning clearer.

While the seasonality is still important in our analysis, the aridity is an even stronger factor.

**L198, L200. Why were these comments about seasonality scrapped?**

L198: This was removed as the mean half flow date cannot be seen as a measure of seasonality.

L200: This was removed as it was based on the assumption that the mean half flow date is a measure of seasonality.

Figure 3. It would be very helpful if the authors add to this caption an example of how to interpret this plot, and how the grey arrows relate to the other information shown. As a side-note, this figure also provides excellent reasons to discuss somewhere the usefulness of fuzzy clustering approaches that avoid imposing strict boundaries on a continuous space. Maybe this could be added to L377-392.

We added an explanation of the arrows to the caption.

Grey arrows indicate the loadings of the original catchment attributes in the PCA space.

We added two sentences in the mentioned part of the manuscript to address fuzzy clustering (see also another reply further below).

It can also be seen that for most of the clusters there is no clear dividing line to neighboring clusters. Therefore, it might be useful to use fuzzy clustering approaches in future research, to avoid those strict boundaries in a continuous space.

L224. ".. are patchier." How does this compare to findings of Addor et al. (2018)?

We added a sentence to address this.

This same pattern can also be seen in some of the signatures used by Addor et al. (2018). Especially the runoff ratio and mean annual discharge form very similar patterns to the clusters in this study.

L236. "..., where the behavior of rivers changes only gradually." Possibly because climate is a dominant driver of hydrology and climate is relatively uniform here?

We also think this is the case and changed the sentence to empathize this.

This also indicates that clustering by using spatial proximity might only work in regions like the eastern US, where the behavior of rivers changes only gradually, due to uniform climate that only changes gradually as well.

L303. "Cluster 5. ..." It would be worthwhile to further emphasize the unique discharge pattern observed in these catchments.

We added a sentence to emphasize this.

They further depict an additional distinct discharge peak in late spring/early summer that separates them from the other catchments found at the west coast.

L344. "However, the frequency ... events is high." It might be helpful to the reader to explain that "high precip" events in the CAMELS database are relative to the mean precipitation, which is very low in these catchments.

We added a sentence to explain this.

However, those high precipitation events are only high in comparison with the mean precipitation for those catchments and not the overall range of precipitation in the entire CAMELS dataset.

Figure 7, Figure 8. To save space and increase readability, perhaps these figures can be combined into a single one.

We kept those two figures separate due to the two column layout of HESS. When the figures are separate they can be shown next to the cluster descriptions, resulting in less scrolling for the reader.

Table 2. The caption is missing an explanation of what "Typical attribute and their manifestation" refers to. It seems very closely related to the "dominating attribute" column. Maybe merge these?

There already is an explanation of this in the caption of Table 2.

Typical signatures/attributes refers to the signature/attribute of the cluster with the lower coefficient of variation scaled by the mean coefficient of variation of the whole dataset. Dominating attribute refers to the catchment attribute that has the highest weighted  $R^2$ .

L408. Section 3.5. It seems that this section has been useful to the authors to describe the hydrologic behavior of their clusters. Should this section perhaps switch places with section 3.4, where the clusters are described? These sections could possibly be merged as well.

We would like to keep the sections in their current order, as we think it is easier to understand when we first explain the hydrological behavior of the catchment and then the influence of the catchment attributes on this behavior.

L528. "Therefore, it is ... few outliers (e.g. Burn, 1997)." This might be another good place to advocate for fuzzy clustering/classification because the catchment space is heterogeneous and continuous and therefor imposing strict boundaries in the hope of finding homogenous groups is unlikely to be successful.

We added a sentence to discuss this.

This hints that for future research a fuzzy clustering approaches might provide less ambiguous results, as it respects the continuous nature of hydrological behavior.

**Textual etc**

176. "However they also ... soil and geology." This sentence has become disconnected from the initial mention of Yeager et al (L170). Change to "However, Yaeger et al. (2012) also ..."

**Changed as proposed.**

Figure 4. This figure seems to have been cropped too tightly along the top and bottom edges. I can't find cluster 5 on it either.

Fixed.

Figure 6. "... approach described (Massmann, 2019)" > change to "... approach described in Massmann (2019)"

Changed as proposed.

457. "(e.g. (Berghuijs ... et al., 2017))" > remove double brackets. Also on lines L474 (twice), L475, L529, L554, L575.

Changed as proposed.

501. "Compared to the ... can be found." I understand what this sentence wants to say but it doesn't quite work. Maybe "The results of this study show some similarities with the clustering results of Kuentz et al. (2017), who derived their cluster from European catchments by an analogous method."

Changed as proposed.

522. "heat" > probably more accurate to change this to "high energy"

Changed as proposed.

534. "create" > "creates"

Changed as proposed.

620. "recmonned" > "recommend"

Changed as proposed.

[revised manuscript text omitted]